# Dendritic Polymers in Tissue Engineering: Contributions of PAMAM, PPI PEG and PEI to Injury Restoration and Bioactive Scaffold Evolution

**DOI:** 10.3390/pharmaceutics15020524

**Published:** 2023-02-04

**Authors:** Michael Arkas, Michail Vardavoulias, Georgia Kythreoti, Dimitrios A. Giannakoudakis

**Affiliations:** 1Institute of Nanoscience Nanotechnology, NCSR “Demokritos”, Patriarchou Gregoriou Street, 15310 Athens, Greece; 2PYROGENESIS S.A., Technological Park 1, Athinon Avenue, 19500 Attica, Greece; 3Department of Chemistry, Aristotle University of Thessaloniki, 54124 Thessaloniki, Greece

**Keywords:** dendrimer, biomaterial, cell scaffold, hydrogel, wound repair, adhesion, differentiation, osseointegration

## Abstract

The capability of radially polymerized bio-dendrimers and hyperbranched polymers for medical applications is well established. Perhaps the most important implementations are those that involve interactions with the regenerative mechanisms of cells. In general, they are non-toxic or exhibit very low toxicity. Thus, they allow unhindered and, in many cases, faster cell proliferation, a property that renders them ideal materials for tissue engineering scaffolds. Their resemblance to proteins permits the synthesis of derivatives that mimic collagen and elastin or are capable of biomimetic hydroxy apatite production. Due to their distinctive architecture (core, internal branches, terminal groups), dendritic polymers may play many roles. The internal cavities may host cell differentiation genes and antimicrobial protection drugs. Suitable terminal groups may modify the surface chemistry of cells and modulate the external membrane charge promoting cell adhesion and tissue assembly. They may also induce polymer cross-linking for healing implementation in the eyes, skin, and internal organ wounds. The review highlights all the different categories of hard and soft tissues that may be remediated with their contribution. The reader will also be exposed to the incorporation of methods for establishment of biomaterials, functionalization strategies, and the synthetic paths for organizing assemblies from biocompatible building blocks and natural metabolites.

## 1. Introduction

Inspiration for the synthesis of polymers with tree-reminiscent morphology was derived from corresponding natural patterns. In the microcosm, these conformations are observed in the fractals, neurons, dendritic cells, veins, and arteries of the blood circulatory system. The roots, branches of the trees, and the river deltas are examples encountered at the macrocosm and are therefore abundant and easily observed [1]. Almost 40 years ago, Buhleier and Vögtle put forward the idea of mimicking these architectures by employing radial polymerization [2]. It was also at this time that D.A. Tomalia gave their first presentation concerning a “Cascade” macromolecule at the Winter Gordon Research Conference, and de Gennes and Hervet first published their work on the structure of a “starburst” [3,4]. Today, dendritic polymers represent the well-established 4th class of polymers, next to their conventional linear, branched, and cross-linked counterparts [5,6,7].

The first reason for this classification resides in their unique structure (Figure 1a). They are constructed by a central core that is the focal point of radial polymerization and may comprise the same monomeric units as the rest of the macromolecule or a completely different entity that endows the substance with exceptional properties. The main body, i.e., the branched interior, contains the monomers and their characteristic groups. These define the conformation of the cavities and their chemical environment. The periphery contains the end groups that may be decorated with functional groups to adapt to the desired role. Accordingly, dendritic polymers are divided into five main categories (Figure 1b). Dendrimers that are perfectly symmetrical and ideally monodisperse (Mw/Mn = 1) [8,9,10,11,12,13] are produced from a tedious procedure of multiple protection-deprotection and purification stages and are costly. For this reason, they have quite limited applicability. Unsymmetrical hyperbranched polymers [14,15,16,17,18,19,20,21] derive from uncontrolled polymerization that results in branches of different lengths. Easy processes lower the overall cost and increase the attractiveness of commercial products. Dendrons [22,23] are fragments of dendrimers or hyperbranched polymers propagating from an active focal point, ideal for surface activation. Dendrigrafts [24,25] show that each monomer of the conventional dendritic branch is replaced by a polymeric chain and dendronized polymers [26,27,28,29,30] are formed by the attachment of dendritic structures to a polymeric core.

The second reason that highlights the importance of dendritic polymers and universal adoption is the multitude of their capabilities [33]. Their most recognized applications reside in the fields of catalysis [34,35,36,37,38], liquid crystals [39,40,41,42], water purification [43,44,45,46], and light harvesting [47,48]. Other implementations are encountered in the areas of textiles [49,50,51], coatings [52,53], separation systems [54,55], chromatography [56,57], membranes [58,59], hydrogels [60,61], solvent extraction [62,63], and color chemistry as nanocarriers of dyes [64,65].

Extensive studies based on molecular modeling have been performed and theoretical predictions have been formulated for the interaction of dendritic polymers with various guests, functionalities, biomolecules, such as genes and proteins, and substrates, such as membranes (Figure 2a) [66]. Their most interesting feature is the similarity to proteins in size, morphology, and chemical environment (Figure 2b) [67]. Due to this advantage, significant advancements have been attained in biomimetic synthesis [68,69,70,71] and biomedical applications in general by employing dendritic macromolecules [72,73,74,75,76,77,78] and their composites [79]. Many dendritic macromolecules have intrinsic antimicrobial [80,81] and antiviral [82,83] properties that are enhanced by adjusting their perimeter [84,85] or incorporating internally reduced metal nanoparticles [86]. Their peculiar “spongy” architecture allows controlled [87], targeted [88], and stimuli-responsive [89] drug delivery [90,91,92]. Further examples deriving from nanomedicine [93,94] include cancer diagnosis [95,96] and therapy [97,98], gene transfection [99,100,101], biosensors [102,103], magnetic resonance and bioimaging [104,105] and theranostics [106,107]. Important research is also performed in other clinical fields, such as the treatment of central nervous system conditions [108,109], therapy for inflammatory diseases, such as rheumatoid arthritis [110,111], antimicrobial coatings for orthopedical implants [112,113], and antigen mimicry for vaccinations [114]. Tissue engineering appears to be the discipline best suited to the properties of bio-dendritic polymers [115,116,117,118].

In the present study, a survey of the methods for dendritic polymers aimed at the artificial restoration of human body functionalities lost due to injury or disease will be attempted. In the first two sections, general concepts will be examined and then tissue-specific cases will be discussed. Gene transfection will be covered only for these particular circumstances. A general analysis of this broad topic is outside the scope of the present review.

## 2. Peptide Functionalized Dendritic Polymers Mimicking Collagen and Elastin

### 2.1. Collagen

Efforts to produce artificial collagen were among the earliest attempts to involve dendritic polymers in tissue engineering. Kinbergert et al. employed a strategy that inspired many alternative implementations, including the binding of amino acid sequences to a central core. Specifically, glycine-proline-N-isobutyl glycine or glycine-N-isobutyl glycine-proline hexamers were attached to a trimesic acid core (Figure 3a), generating “first-generation dendritic polymers” (rather star polymers) [119]. Both derivatives adopted triple helices conformation reminiscent of collagen (Figure 3b), with the Gly-Nleu-Pro sequence analog being more thermally stable. Replacement of the central core with G0.5 poly(amidoamine) (PAMAM) afforded a collagen mimetic dendrimer PAMAMG0.5[(Gly-Pro-Nleu)_6_-OMe]_8_ [120] (Figure 4a) that was more suitable for wound healing and bone mineralization, since it had the potential to readily form complexes with biologically relevant Cu^2+^ and Ni^2+^. Analogous results were obtained by G4 PAMAM and a similar pentamer (Pro-Pro-Gly)_5_ [121], where the triple helix conformation is reformed in contrast to conventional collagen after the thermal transition to gelatin and subsequent cooling. Elongation of the same peptide sequence to decamers improved the similarity of the triple helicity to the standard collagen [122]. Furthermore, the product was capable of temperature-dependent hydrogel formation without a crosslinker. Ultimately in a final modification, the same group replaced the Pro-Pro-Gly sequence with Gly-Pro-Hyp and produced heat-induced, collagen-like hydrogel that does not undergo thermal denaturation to gelatin (Figure 4b) [123]. G1–G3 poly(propylene imine) dendrimers were exploited for the replacement of the cytotoxic glutaraldehyde in the 1-ethyl-3-(3-dimethyl aminopropyl) carbodiimide hydrochloride-induced crosslinking of conventional collagen for the production of hydrogels resistant to proteolysis, which could form films characterized by high biostability [124].

### 2.2. Elastin

The above-described concept of functionalizing the periphery of a dendrimer also applies to the approximation of elastin aggregates. Replacement of the external amino groups of G4 PAMAM by a single peptide sequence: Val-Pro-Gly-Val-Gly, causes the conversion of the conformation from the random coil into b-turn. In addition, the relevant temperature can be adjusted by modifying the pH or salt concentration [125]. It is also dependent on the oligopeptide length and the generation of the dendrimer. Both the dimerization of the former peptide and substitution with G5 PAMAM lower the transition temperature and their effect is synergistic [126]. This decrease in temperature is very important as it reaches body temperature and thus renders the conjugate a temperature-responsive biomaterial. The transformation to b-turn causes dehydration and increases the hydrophobicity of the dendritic derivative, leading to aggregation and the subsequent development of turbidity (Figure 5a). The incorporation of gold nanoparticles into the dendritic cavities affords dual stimuli-sensitive hybrids, both photothermogenic and thermoresponsive [127] (Figure 5b). All the mimetic approaches of collagen and elastin are summarized in Table 1.

## 3. Dendritic Polymers Mediating to Cell Adhesion, Proliferation, and Differentiation

### 3.1. Scaffolds from Collagen and Beyond

Biocompatibility is the most crucial factor for the assessment of a potential biomacromolecule. For this reason, the behavior of key cell types in contact with biologically relevant substrates and scaffolds containing dendritic polymers was thoroughly checked [128]. Fibroblasts are the most common and convenient test subject as they may respond quickly to the transfection initiators included in dendritic polymers, differentiate, and adapt to the desired activity. Beneficially, in its first activity, G1 PAMAM crosslinked with the cholecyst-derived extracellular matrix and stabilized the resulting scaffold against collagenase degradation, while seeded murine 3T3 fibroblasts retained their phenotype morphology proliferation rhythm and metabolic activity [129]. Analogous effects are generated if G2 PAMAM replaces glutaraldehyde as the substantially non-toxic crosslinker, enhancing the structural integrity of collagen and the resistance to enzymes. The proliferation of human conjunctival fibroblasts after 3 days of incubation increased by 230% in comparison to the control and by about 30 in comparison to the sample without dendrimer [130]. In a similar implementation, collagen-G1 PAMAM crosslinked additionally with three pro-survival peptide analogs: BMP2 mimetic peptide (5hexynoic-KIPKASSVPTELSAISTLYL), EPO mimetic peptide (5hexynoic-GGTYSCHFGPLTWVCKPQGG, disulfide:C6–C15) and FGF2 mimetic peptide (5hexynoic-YRSRKYSSWYVALKRK(YRSRKYSSWYVALKR)-Ahx-Ahx-AhxRKRLDRIAR-NH_2_) (Figure 6a). The resulting scaffold protracted the release of these growth factors and the survival of therapeutic stem cells in the ischemic microenvironment of SCID Beige mice [131]. G2 poly (propylene imine) PPI dendrimer presents an additional alternative for the manufacturing of crosslinked matrices with commercial bovine type-I collagen, human collagen, and human extracellular-based products (hECM) suitable for the unhindered evolution of 3T3 fibroblast colonies [132]. Research in this area has advanced to the point of the development of a commercially available activated G6 PAMAM transfection reagent: SuperfectTM. This in turn, combined with a collagen-chondroitin sulfate-based scaffold, was used for the successful differentiation of mesenchymal stem cells [133].

Although collagen-derived templates are very common and convenient for cell cultivation and tissue evolution, some other solutions have been proposed. Silk fibroin is a protein that under appropriate treatment undergoes a transition from random coil to b-sheet. Electrospinning of lyophilized sponges yielded carboxy-group-rich nanofibers, an ideal substrate for interaction with the amino groups of G2 PAMAM. MTT assay revealed improved affinity to fibroblast cell line L929 in terms of adhesion and proliferation (10% increase after 24 h) [134]. Poly-lactide (PLA) filaments represent a different suitable material for 3D-printing substrates for cell growth. Fluorescent rhodamine G5 PAMAM derivative coatings (Figure 6b) not only facilitate the human HeLa cells approach and enhance their proliferation (320 and 450%, after 2 and 3 days of incubation respectively instead of 150 and 250% of the untreated samples) but are also able to induce their differentiation by transfecting miRNA mimic oligonucleotides such as premir-503 [135]. Gelatin also presents practicable features. In dual functionality as a tissue engineering scaffold and, in parallel, as a sustained drug delivery vessel, gelatin methacrylate (GelMA) was combined with G2 PAMAM in the form of cryogel doped with the anti-inflammatory corticosteroid betamethasone sodium phosphate (BSP). In comparison to cellulose nanocrystal carrier (CNC), PAMAM excelled in augmenting the loading capacity protracting the degradation time of the hydrogel matrix and the release of the active ingredient (Figure 6c). Cytocompatibility was established against the L929 cell line by MTT assay after 48 h of culture and 4 h of incubation. None of the cryogel samples exhibited toxicity, while in contrast, the cell proliferation rate was faster in comparison to the control tissue culture plate [136].

**Figure 6 pharmaceutics-15-00524-f006:**
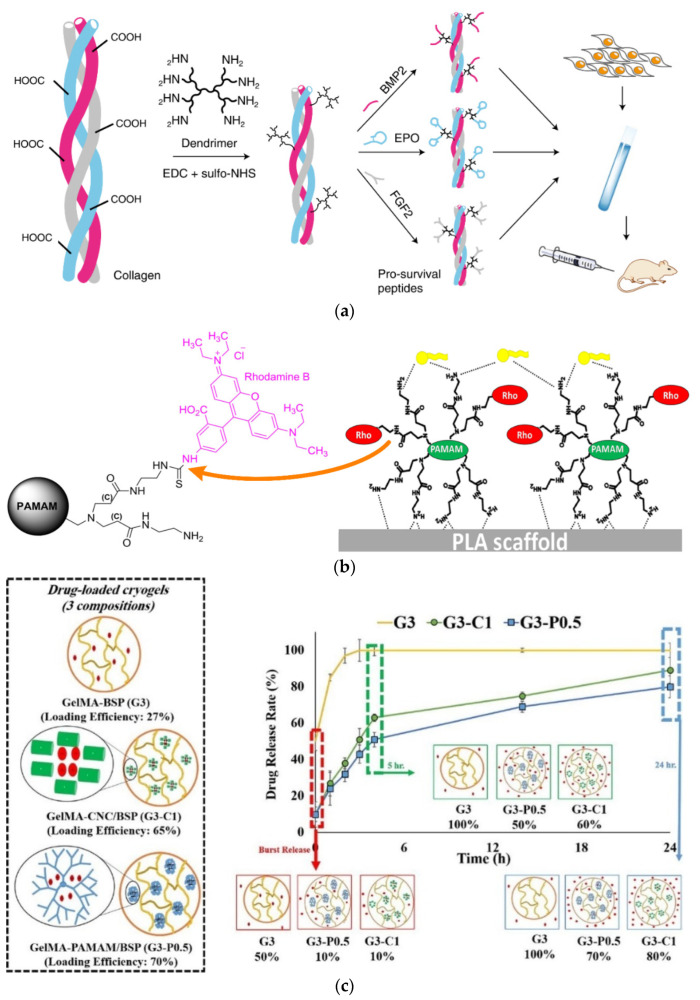
(**a**) Schematic depiction of the method of collagen-G1 PAMAM-peptide synthesis. Reproduced with permission from [131]. Copyright: The Author(s); (**b**) Rhodamine-PAMAM bioconjugate and schematic representation of the transfection process by premir-503 loaded dendrimers layered on the PLA surface. Adapted under permission from [135]. (**c**) Betamethasone sodium phosphate (BSP) loading efficiency to scaffolds of 3 different compositions: gelatin methacrylate (GelMA); GelMA and cellulose nanocrystal carrier (CNC); GelMA and PAMAM; together with the corresponding release profiles. Under permission from [136] Copyright: Elsevier Ltd.

### 3.2. Hydrogel Matrices

Different platforms have been proposed beyond the typical solid scaffold composition. Among them, perhaps the most popular is hydrogels [137,138]. In general, there are three crosslinking strategies for assembling hydrogels with dendritic polymers. Dendritic macromolecules may (a) undergo self-crosslinking with the aid of conventional crosslinkers or their terminal functional cross-linking groups, (b) conjugate with another usually linear biopolymer, such as PEG, and (c) form dendron-linear-dendron entities that may follow any of the two previous crosslinking paths (Figure 7) [139].

Photoreactive hydrogels containing G3, G3.5 PAMAM and PEG-acrylate of variable length (MW 1500, 6000, or 12,000) were prepared by UV-curing and the aid of an eosin Y-based photoinitiator. They exhibited good cytocompatibility towards RAW264.7 mouse macrophages and the potential to fulfill the dual role as both hydrophilic and hydrophobic drug carriers (Figure 8a) [140]. Another similar example of acrylate moieties leading to photo cross-linkable hydrogels concerns hyperbranched polyesters (HPE) self-gelation by UV irradiation. In addition to the sustained release of hydrophobic drugs such as dexamethasone acetate, controlled adhesion, proliferation, and spreading of NIH-3T3 fibroblast-like cells was attained by fine-tuning the gel formulation (Figure 8b) [141]. Furthermore, radiation dendritic polymer cross-linking may be induced by enzymes. Horseradish peroxidase (HRP) is a successful example employed for the oxidative gelation of phenol-terminated hyperbranched polyglycerol (PG) (Figure 9a). The reaction is mediated by gluco-δ-lactone and may be triggered by the glucose in human blood or wound exudates (Figure 9b). Testing by murine line L929 fibroblast cells revealed that they may be readily encapsulated in the gel matrix [142]. A very recent study published in 2023 also describes microspheres of hydrogels from Poly(L-lysine) dendrigrafts and PEG-bis(N-succinimidyl succinate) that can enhance osteogenic differentiation of Bone Marrow Mesenchymal Stem Cells [143].

### 3.3. Dendritic Glues for Cell Aggregation

Most of the common dendrimeric molecules (PAMAM, PPI, poly(ethylene imine) (PEI)) form nitrogen cations. For this reason, they have an inherent affinity to adhere to cell surfaces by electrostatic counterbalancing their negative charge surplus. The ability was further exploited to promote regulated cellular aggregation. A first intercellular linker was synthesized from G3 PPI by attachment of oleic acid moieties to the dendritic periphery through polyethylene glycol spacers. The anchorage of hydrophobic oleyl groups to the cell double-layered aliphatic chain membrane produced a synergistic effect with electrostatic neutralization, promoting cell clustering. Mechanical confinement by centrifugation contributed to the controlled ordering of multi-C3A cell cellular assemblies into defined three-dimensional morphology, shaping cellular rods, rings, and sheets that retained the high levels of viability and proliferation observed with simple conventional complexes with dendrimers [144]. Three-dimensional multi-cellular assemblies (MCA) were observed from aldehyde-modified HepG2 hepatoma cells with the mediation of hydrazine-substituted PPI intracellular linkers. G3 and G4 analogs exhibited the best results in terms of cytotoxicity and cell proliferation. Cellular functionalities, i.e., albumin secretion and cytochrome P-450 activity, were improved in comparison to the 2D monolayer cultures, while the spherical morphology of the cells was retained as depicted in Figure 10A. From the same series of pictures on cellular aggregate formation, it can be deduced that 3D cell interactions and inter-cellular binding are rather loose during the first day and gradually become more and more compact [145]. A similar effect was reproduced by the modification of G4 PAMAM by an adhesive RGD oligopeptide (cysteine-glycine-arginine-glycine-aspartic acid-serine), resulting in accumulations from NIH 3T3 fibroblast spheroid cells (Figure 10B) [146]. The same amino acid sequence is a ligand of integrin typically expressed by hepatic cells. Bound to PAMAM via a polyethylene glycol spacer promotes the aggregation of liver cells and their ammonia metabolism function [147]. Collagen-mimetic dendrimer derivatives may act as cell aggregation moderators too. To attain the triple helix conformation, two modified versions of G1.5 PAMAM-COOH were enzymatically crosslinked by tissue transglutaminase. The first contained a cell-binding peptide sequence (GFOGER) and an amine donor substrate: residues 2800–2807 from human fibrillin-1 (EDGFFKI). In the second, the donor was replaced by an amine acceptor probe derived from human osteonectin (APQQEA) (Figure 10C). The conjugated product was proven to be an integrin-specific Hep3B cell adhesion factor (Figure 10D) [148]. All the proposed methods for substrate bioactivation and cell organization are summarized in Table 2.

## 4. Bone Restoration

To follow fluently the advancements towards bone restoration, a comprehensive description is quoted. The hierarchical structure of long hollow bone presents many similarities with bamboo (Figure 11), reflecting nature’s tendency to preserve and evolute the most successful patterns. Three chains of collagen peptides are entwined to form Type-I collagen triple helix protein (tropocollagen), which in turn is mineralized by hydroxyapatite nanocrystals to form collagen fibrils (diameter ~1.5 nm, length ~300 nm). Several fibrils are in turn organized in fibers that are arranged in lamellae. Concentric layers (8–15) of the latter around a central Haversian canal that contains the blood vessels and supplies blood to the bone constitute the osteons. This compact/cortical part engulfs a spongy/trabecular bone foam to complete the osseous architecture. In bamboo, the fundamental unit is replaced by cellulose, which is organized in microfibrils and then in fibers. These are incorporated in lignin–hemicellulose hollow-shaped cells that eventually form the bamboo bark [149].

### 4.1. Hydroxy Apatite Nucleation and Functionalization

Biological apatite nucleation enjoyed thorough research lately with state-of-the-art imaging, for instance, computer-aided three-dimensional visualizations of tomograms and structure determination techniques such as high-resolution transition electron microscopy (TEM) and cryo-TEM. In both bones and teeth, the existence of an amorphous calcium phosphate precursor phase deriving from calcium triphosphate prenucleation clusters has been identified before the generation of apatite crystals [150,151,152,153] (Figure 12A). Even though hydroxyapatite (HAP) is the most important ingredient of all major hard tissues the first indications of the bioinspired capability of dendritic polymers to promote its formation did not come from the research on tissue engineering. Khopade and his group initially reported the crystalline modification of calcium phosphates by G3.5 and G4.5 PAMAM-COOH and the templating ability of the latter to shape spherical hydroxyapatite nanoparticles. Moreover, they recommend two possible nucleation mechanisms (Figure 12B). Coating with a sugar layer and loading with hemoglobin afforded oxygen-carrying aquasomes [154]. Soon after, biomimetic HAP development by a combination of PPI and three different surfactants (octadecyl amine, hexadecyl trimethyl ammonium bromide, and sodium dodecyl sulfate) was associated with bone remodeling, introducing higher stiffness and toughening [155]. Oxidized multi-walled carbon nanotubes were also bioactivated with G2 and G3 PPI and were used to form nanostructured HAP by the wet precipitation method. The resulting composites exhibited supplementary anticancer activity as established by human osteosarcoma (MG-63) cells [156].

In modified PAMAM dendrimers with dodecane diamine core, the terminal groups were replaced with succinamic acid. Gold nanoparticles spontaneously accumulated into the cavities without an additional reducing agent. The final composite was applied on poly(2-hydroxyethyl methacrylate) and proved capable of biomimetic hydroxyapatite calcification in synthetic body fluid [157]. The molecules of an amphiphilic Newkome-type dendron bearing cholesterol as the lipophilic moiety, orient at the air–liquid interface and form monomolecular films-templates. To these matrices, by increasing the concentration of calcium and phosphate ions, octacalcium phosphate and HAP needles, spheres, and flakes form due to the mediation of the dendritic hydrophilic segments (Figure 12C) [158].

Hydrothermal treatment in synergy with the dendritic matrices proved very beneficial for the orchestration of HAP crystallization. Nanorods were formed by calcium and phosphate ions incubation at 150 °C with G5.5 PAMAM-COOH [159] and G2.5 PAMAM-COOH, as well as with a polyhydroxylated analog [160]. At the same temperature, generation G3 and G4 PAMAM led to ellipsoid-like nanocomposites [161]. G4 PPI was employed for an easy procedure involving different concentrations and optional hydrothermal treatment for controlled accumulation of HAP in terms of size distribution and morphology. The dendritic matrix favored HAP crystallization in comparison to the other phases of calcium phosphate, even at an ambient temperature [162]. Employing the same method, hydrothermally nucleated HAP nanocrystallites templated by hyperbranched PEI were blended with chitosan or N-acetylated chitosan to construct porous composite scaffolds. The higher mechanical strength observed for the latter was attributed to an extended hydrogen bond network among neighboring polymeric chains [163]. The blending of a dendritic polymer with chitosan formulation is adopted by researchers in the area of hydrogel formation. This time, chitosan was covalently grafted to hyperbranched G1.5, G2.5, and G3.5 PAMAM, and then used for nanocrystalline HAP nucleation. Gelification followed, with chitosan using genipin as a crosslinker and the resulting hydrogels were doped with anti-inflammatory ketoprofen to put to use the capabilities of the hyperbranched polymer for controlled release [164].

The potential of hyperbranched PEI to organize spherical Ca_3_(PO_4_)_2_ to hybrid nanoparticles [165] was exploited to develop biomimetically bioactive layers in inert substrates. The first implementation involved titanium, the basic material of orthopedical implants. After polishing, chemical pretreatment, and an intermediate triethoxy chloropropyl propyl silane functionalization, the hyperbranched polymer PEI matrices were covalently bonded to the metal substrate (Figure 12D). The calcium phosphate layer was then deposited hydrothermally [166]. In an advancement of the previous technique, G4 PPI mediated HAP nucleation and subsequent hydrothermal treatment of Ti coupons, with the resulting dispersion affording the stable coating of HAP nanorods [167].

**Figure 12 pharmaceutics-15-00524-f012:**
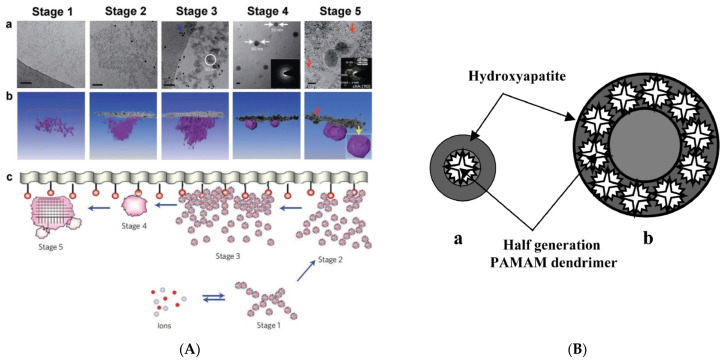
(**A**) (**a**) Two-dimensional cryo-TEM images of different stages of surface-induced apatite nucleation from simulated body fluid. Scale bar = 50 nm. (**b**) Computer-aided three-dimensional visualizations of tomograms. The inset Selected area (electron) diffraction (SAED) at stage 4 shows the amorphous spherical particles attached to the monolayer. The inset SAED at stage 5 can be indexed as carbonated HAP with a [110] zone axis. (**c**) Scheme of the mineralization process Stage 1: loose aggregation of prenucleation clusters in equilibrium with ions in solution. Stage 2: prenucleation clusters aggregate in the presence of the monolayer with loose aggregates still present in the solution. Stage 3: aggregation leads to densification near the monolayer. Stage 4: nucleation of amorphous spherical particles only at the monolayer surface. Stage 5: development of crystallinity following the oriented nucleation directed by the monolayer. Reproduced with permission from [150] Copyright: Nature Publishing Group and [168]. Copyright: Nature Publishing Group. (**B**) Possible mechanisms of slow hydroxyapatite growth. (**a**) Dendrimer as a nucleus for hydroxyapatite precipitation and growth. (**b**) Dendrimer supramolecular aggregate as a nucleus for hydroxyapatite precipitation and growth. Reproduced with permission from [154]. Copyright: Elsevier Science B.V. (**C**) Structure of the Cholesteryl Newkome-type dendron amphiphile and different Ca_3_(PO_4_)_2_ crystal shapes as a function of nucleation time. Reproduced with permission from [158] Copyright: WILEY-VCH Verlag GmbH & Co. KGaA, Weinheim. (**D**) Schematic representation of the reaction steps carried out for the development of a chemically attached hyperbranched polymeric layer to alkali-treated Ti surface Reproduced with permission from [166]. Copyright: The Author(s).

The chemical environment of the dendritic periphery plays a crucial role in the morphology of the deriving HAP crystals. Modification of the G2 PAMAM end groups by glutamic acid allowed the growth of tape-like crystals of HAP and octacalcium phosphate in single and double diffusion systems of agarose or gelatin [169]. Tri-phosphate (DPG3.0-P3) or bis-phosphonate (DPG3.0-P2) terminal groups attached to dendronized poly(amido amine)s (Figure 13a) direct the nucleation of ‘‘fish-bone’’-like or fibrous HAP nanocrystals. On the microscale, they aggregate into ribbons or fibrils well suited for the adhesion and proliferation of bone marrow stromal cells (Figure 13b) [170].

There are two last examples of nonconventional dendritic polymer connections with HAP. The first concerns the use of G4 PAMAM or cetyltrimethylammonium bromide as porogens for the production of mesostructured HAP. Two Mechanisms were described: (a) quaternary ammonium salt micelles create porosity that leads to mesoporous HAP, which is subsequently functionalized with the dendrimers (Figure 14, Scheme A; the porosity may be fine-tuned by modifying the quaternary ammonium salts [171], (b) pores are generated by PAMAM clusters and HAP is produced as a result of calcination (Figure 14, Scheme B) [172]. The second again concerns surface-tailored HAP by G1 to G5 PAMAM dendrons possessing a triethoxy silyl focal point (Figure 14b) [151].

### 4.2. Osseointegration

#### 4.2.1. Adhesion and Proliferation of Osteoblasts and Other Cells Related to Bone Development

One of the critical challenges encountered in most artificial bone implants is the incompatibility of artificial materials with natural bone. The most common material for orthopedical implants, titanium, was the obvious choice for a comparative study of its interaction with dendritic substances. G5 PAMAM, with terminal amino groups substituted (0–60%) with PEG chains, exhibited high affinity and strong adsorption. Stable films were formed on bare metal surfaces and analogs coated with calcium phosphates through micro-arc oxidation. Potent antibacterial activity was recorded against Gram-negative *Pseudomonas aeruginosa* and weaker against Gram-positive Staphylococcus aureus. Furthermore, human bone mesenchymal stem cell proliferation was not affected [174]. Many efforts have been made to improve the attachment of osteoblasts and other related cells. The first suggestion was an organic-inorganic biodegradable hybrid film comprising PEG-Dendritic Poly(L-lysine) Star Polycaprolactone and commercial HAP that provided a suitable substrate for the immobilization of MG63 osteoblast-like cells. After 24 h of incubation, cell adhesion increased substantially in comparison to the polystyrene control [175]. Alteration of the peripheral charges of G5 PAMAM covalently bound to a silicon surface was performed by changing external amino groups with methyl or carboxyl. The attraction of human MG63 osteoblasts to positive charges of the ammonium cations was established and the production of organized actin stress fibers was observed (Figure 15a) [176]. The famous Arginine-glycine-aspartic acid RGD tripeptide was employed to decorate G1 PAMAM to enhance the immobilization of mesenchymal stem cells on a polystyrene substrate in an amino acid sequence-specific way [177]. Analogous amide-based amino terminal dendrons chemically bonded to titanium implants (Figure 15b) reproduced the same effect for human osteoblastic cells [178]. Bifunctional dendrimers were needed in the inert gold surfaces to combine firm attachment to the metal substrate and good adaptation to the cell’s requirements. Thioctic acid was bonded to cyclotriphosphazene nucleus for the immobilization of the organic layer, whereas carboxy terminated dendritic branches based on poly(phosphorhydrazone) propagated from the same core (Figure 15c) for the adhesion and proliferation of human osteoblasts [179]. A 10% increase in cell viability was observed after 280 h for the positively charged dendrimers.

Hydrogel templates evolved next to coatings. The arginyl glycyl aspartic acid (RGD) oligopeptide was incorporated into a CGRGDS peptide sequence and was bound to a G5 PAMAM. Blending with thiolated hyaluronic acid (HS-HA) produced a hydrogel matrix suitable for the adhesion and proliferation of bone marrow stem cells. RGD-modified PAMAM doubled the cell’s metabolic rate after 24 h [180]. Furthermore, cross-linking sodium alginate with G1 PAMAM produces an injectable hybrid cationic hydrogel platform for the electrostatic immobilization of negatively charged MC3T3-E1 pre-osteoblasts and their successful cultivation [181].

For the manufacture of three-dimensional electrically conductive biocompatible scaffolds, the fourth generation of a hyperbranched aliphatic polyester was synthesized by melting polycondensation of tris(methylol)propane (TMP) with 2,2-bis(methylol)propionic acid. Subsequently, it was peripherally decorated with thiophene moieties. These groups underwent polymerization, yielding star-shaped polythiophene with a hyperbranched core. Electrospinning with poly(ε-caprolactone) afforded hydrophilic conductive and biocompatible nanofibers that permitted the attachment and rapid proliferation of the mouse osteoblast MC3T3-E1 cell line (Figure 15d) [182]. After 7 days, an 8-fold cell expansion was observed for the electrospun nanofibers and 7.2 on the control culture. On the other hand, composite scaffolds containing carbon nanotubes, G3 PAMAM, and in situ nucleated nanostructured HAP favored the proliferation of the osteoblast-like MG 63 cell line in comparison to their G1 and G2 counterparts [183].

**Figure 15 pharmaceutics-15-00524-f015:**
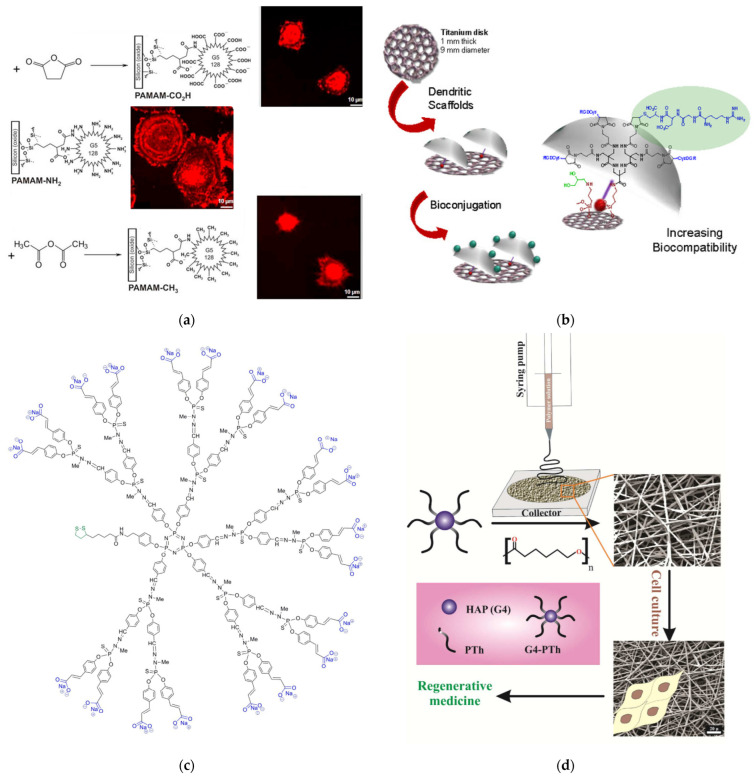
(**a**) Synthesis of PAMAM surface coatings via G5-dendrimers on Si oxide bulk materials with terminal amino (NH_2_), carboxylic acid (CO_2_H), and methyl (CH_3_) groups and confocal images of human MG-63 cells growth after 1 h cultivation. Reproduced with permission from [176]. Copyright: Elsevier B.V. (**b**) Schematic representation of the biofunctionalization of titanium implants reproduced under creative commons license from [178]. (**c**) Chemical formula of water-soluble bifunctional dendrimer having one dithiolane function at the core, poly(phosphorhydrazone) branches, and carboxy-terminal groups. Reproduced under creative commons license from [179]. (**d**) The overall methodology for the fabrication of biocompatible, porous, and electrically conductive scaffold for regenerative medicine. Reproduced with permission from [182]. Copyright: Wiley Periodicals, Inc.

#### 4.2.2. Cell Differentiation

As mentioned in the introduction, extensive research has been performed on the potential of dendritic polymers for gene delivery. This property has been exploited in the field of osteogenesis. Mesenchymal stem cells were the appropriate candidate hosts, since they may readily proliferate and differentiate into a multiplicity of cell types, among them osteoblasts. G5-G7 PAMAM is reported as a vector that delivers the β-galactosidase or human bone morphogenetic protein-2 gene to differentiate rat bone marrow mesenchymal stem cells towards osteoblasts [184]. Transfection efficiency was generation-dependent and gene expression was generally low but adequate to promote the genetic modification of the cells. The efficacity of gene delivery was improved in a second more sophisticated approach, where PAMAM dendrimers were coupled with histidine and arginine or lysine or ornithine (Figure 16a). Osteogenic, adipogenic, and chondrogenic differentiation to human adipose-derived mesenchymal stem cells was accomplished [185]. The conjugation of G5 PAMAM with 3 different aminoacid sequences: Arg-Gly-Asp (RGD), Tyr-Ile-Gly-Ser-Arg (YIGSR), and Ile-Lys-Val-Ala-Val (IKVAV), also contributed to the adhesion of bone marrow mesenchymal stem cells [186].

The property of dexamethasone (Dex) to control the proliferation and differentiation of stem cells has been joined with the capacity of PAMAM to host active ingredients and deliver them in an orderly fashion. Carboxymethyl chitosan terminal groups to G1.5 PAMAM-COOH cause aggregation to nanospheres that incorporate Dex. Internalization of the complex into rat bone marrow stromal cells induced modification toward an osteoblastic phenotype [187]. An analogous result was observed by the loading of the composite nanospheres onto HAP and starch polycaprolactone (SPCL) scaffolds. Furthermore, this addition amplified cell adhesion and proliferation in comparison to the untreated scaffolds [188]. The treated HAP macroporous ceramics [189] and starch-polycaprolactone organic matrices [190] (Figure 16b) were subcutaneously implanted on the back of rats and caused the production of the proteoglycan extracellular matrix and ectopic bone formation.

**Figure 16 pharmaceutics-15-00524-f016:**
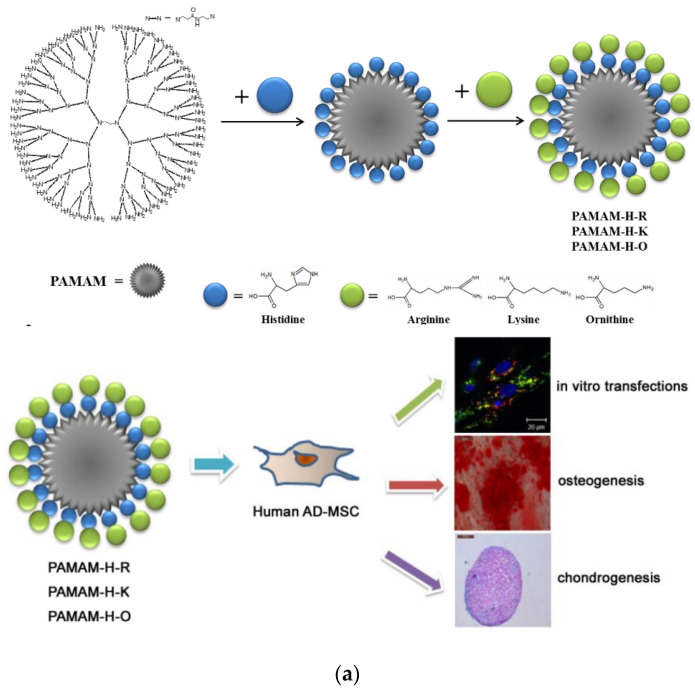
(**a**) Schematic illustration of the synthesis of conjugated PAMAM derivatives and their effects on the differentiation of human adipose-derived mesenchymal stem cells. Reproduced with permission from [185]. Copyright: Elsevier B.V. (**b**) Schematic illustration of ex vivo stem cells “tune-up” and de novo bone tissue formation in vivo. Dendronlike NPs are internalized by stem cells in vitro, and then cells are seeded onto starch-polycaprolactone scaffolds and implanted. Release of dexamethasone to the cytoplasm will occur and de novo bone formation (blue areas) is observed within SPCL scaffolds after 4 weeks of implantation. Reproduced with permission from [190]. Copyright: Elsevier Inc.

Substantial studies have been carried out on the osteoconductive ability of a particular type of G3 poly(1-lysine) dendrons peripherally modified by phosphoserine. The latter is a well-recognized promoter of calcium phosphate biomineralization and proved capable of forming a homogeneous bioceramic layer, accelerating the proliferation of MG63 and SAOS-2 osteoblast cells and facilitating differentiation [191]. Further testing on calvaria-derived MC3T3 osteoblastic cells, mesenchymal, undifferentiated C2C12 cells, and primary bone marrow cells confirmed differentiation towards osteoblastic phenotypes by the detection of high levels of alkaline phosphatase, osteocalcin, and osteoprotegerin [192]. The same phosphoserine-tethered hyperbranched molecules were incorporated in calcium phosphate gels to amplify the proliferation and differentiation of mesenchymal stem cells [193].

The strategy of dendritic polymer functionalization with aminoacid sequences to secure cytocompatibility was carried out from simple coatings to scaffolds. The blending of a linear copolymer of poly(lactic acid)-b-poly(ethylene glycol)(PEG)-b-poly(lactic acid) with acrylate end-groups (PLA), together with G4.0 PAMAM with polyethylene glycol-arginine–glycine–aspartic acid–D-tyrosine–cysteine(RGDyC)-acryloyl terminal groups and subsequent photo-cross-linking afforded hydrogel matrices with elevated porosity, low swelling, and mechanical stability (Figure 17a). Experiments with mouse bone marrow mesenchymal stem cells highlighted their exceptional adhesion, differentiation, and proliferation, as well as the potential of this platform for bone formation [194]. Crosslinking a divinyl sulfone G5 PAMAM derivative decorated with the fibronectin RGD peptide with an eight-armed thiolated polyethylene glycol produced one more formula for osteogenic hydrogels, promoting the differentiation of mesenchymal stem cells. The latter was favored instead of adipogenic differentiation in the higher concentrations of the dendritic component [195]. In a contiguous mechanism, G5 PAMAM was decorated by an Fc-binding polypeptide (Figure 17b). Together with thiolated hyaluronic acid (HS-HA), it formed hydrogel that was capable of promoting the immobilization and development of human umbilical cord mesenchymal stem cells. Furthermore, it up-regulated the expression of the human vascular endothelial cadherin fusion protein, which in turn is very important for vascularization in tissue engineering (Figure 17c) [196]. Aside from hydrogels, multiwalled carbon nanotube matrices covalently bonded with G3 PAMAM were employed for the mineralization of strontium-substituted HAP and the protracted release of osteoconductive curcumin. The obtained hybrid hydrophilic scaffold was mechanically robust and promoted osteoblast-like MG-63 cell proliferation and differentiation (Figure 17d) [197].

### 4.3. Bone Formation

The pioneering work of bone formation onto ceramic HAP scaffolds loaded by dexamethasone/carboxymethyl chitosan-PAMAM, described in the previous study [189], was followed by a similar in vivo study. Scaffolds made of chitosan and nano-hydroxyapatite formed biomimetically with the aid of hyperbranched PEI were introduced in Sprague Dawley rats with calvarial critical-sized defects. Guided bone regeneration and a large number of osteocytes were observed [198,199]. Newly formed bone was detected by histological analysis at the lateral area inward of the middle sagittal seam and the lateral area outward of the middle sagittal seam (4338.5/4340.0 μm^2^ and 17,321.5/12,123.0 μm^2^ after 2 and 4 weeks, respectively. The capability of phosphoserine functionalized poly(epsilon-lysine) dendrons to orient, immobilize and develop osteoblast cells, as well as to promote their bone formation potential was reasonably implemented in vivo experiments. Coated titanium implants were embedded in the pelvic bone of sheep and assisted osseointegration and rapid bone regeneration [200]. However, these results were not reproduced in ZirTi implants introduced in the right side of the mandible of Beagle dogs. Osseointegration proceeded smoothly in the bare and the phosphoserine/polylysine dendrimer-coated inserts but no rate difference was monitored [201]. Mixing with strontium-doped hydroxyapatite gels [202] or biphasic Ca_3_(PO_4_)_2_-Sr_3_(PO_4_)_2_ injectable bone pastes (Figure 18a) reduced chronic inflammatory response from macrophages, while implantation in osteoporotic rats proved their potential for bone healing (Figure 18b) [203] and the proliferation and differentiation of mesenchymal stem cells. After 8 weeks, histological analysis and micro-computed tomography revealed that bone formation is doubled when strontium-doped hydroxyapatite is combined with G3 hyperbranched poly(epsilon-lysine) dendrons, exposing phosphoserine. A much more elaborate hydrogel produced from G4.0 PAMAM bonded with an arginine-glycine-(aspartic acid)-(D-tyrosine)-cysteine (RGDyC) peptide through PEG chain spacers and an 8-arm PEG star polymer with 3, 4-dihydroxy-L-phenylalanine terminal groups, implanted to a mouse with calvarial critical size defect, caused a modest formation of “bone-like” tissue and substantial “bony” tissue (Figure 18c) [204]. The scientific work dedicated to the treatment of damage to the bones is summarized in Table 3.

## 5. Applications of Dendritic Polymers in Dentistry

### 5.1. Interactions with Odontoblasts and Dental Pulp Cells

Teeth are the second most abundant hard tissue of the human body next to bones Although hydroxyapatite is the most abundant component of their major constituent’s dentine and enamel [205], research in this field has followed an inverse direction, focusing instead on materials that interact with cells and inorganic component reconstitution. A possible explanation is that, unlike the bone, for the formation of enamel, the tooth does not have the cell equivalent to the osteoblast that forms the organic matrix and mineralizes it. In the case of teeth, the ameloblast dies with the eruption of the tooth.

In this approach, G5 PAMAM was functionalized by peptides incorporating the typical RGD cell adhesion sequence (Arg-Gly-Asp). The α_ν_*β*_3_ integrin binding potential was expressed for human dermal microvessel endothelial cells (HDMEC), human vascular endothelial cells (HUVEC), and odontoblast-like MDPC-23 cells. However, the most important aspect of this work was the targeting capability of these compounds to the RGD receptors of the predentin of human tooth cultures, rendering them appealing carriers for tissue-specific cell delivery [206]. Similar G5 PAMAM RGD peptide composites synthesized by the intervention of fluorescein isothiocyanate also demonstrated a selective binding capacity to dental pulp cells and mouse odontoblast-like cells. On top of that, they modulated their differentiation toward the improvement of their odontogenic properties [207]. PAMAM derivatives phosphorylated via the Mannich-type reaction (Figure 19) may also mediate odontogenic differentiation of the dental pulp stem cells and assist their proliferation [208].

### 5.2. Dentin Reconstitution

The specific investigation into the contribution of dendritic polymers to the biomineralization of teeth components (Figure 20a,b) began with a time gap of about one decade in comparison to the bones [209]. Established agents, such as the carboxy-terminated PAMAMs, were employed in a biomimetic attempt to imitate the role of non-collagenous proteins in the hierarchical intrafibrillar mineralization of dentine (Figure 20b). Both in vitro and in vivo experiments in mice highlighted the effectiveness of G4 PAMAM-COOH in the sequestration of calcium and phosphate ions and the intrafibrillar templating of hydroxyapatite from amorphous calcium phosphate (ACP) (Figure 21a) [210]. The results were confirmed in vivo for G3.5 PAMAM-COOH and incubation in mice saliva after incorporation into the rat’s cheeks. On the side, the produced hydroxyapatite exhibited increased microhardness [211].

It seems that biomimicry of PAMAM is independent of generation and terminal groups. The above results were reproduced by G3 PAMAM-NH_2_ [212] or G3 PAMAM-NH_2_ combined with glutaraldehyde (Figure 21b) [213,214], or G4 PAMAM combined with a peptide bond condensing agent [215] in terms of dentine permeability [216] calcium absorption, adhesion to the demineralized dentin samples immersed in artificial saliva, and the production of needle-like hydroxyapatite crystals. Furthermore, G4 PAMAM-NH_2_, G2 PAMAM-OH, and G4 PAMAM-OH caused occlusion of the dentinal tubules that could withstand an acid environment (Figure 21c) [216,217]. A comparative study of generation 3 PAMAM-NH_2_, PAMAM-COOH, and PAMAM-OH revealed the superiority of the first two derivatives concerning decreased lesion depth and mineral loss and increased mineralization rate of dentin tubules, dentin hardness, and percentages of Calcium and Phosphorus [218]. Besides dentin, the same biomimetic mechanism of intrafibrillar mineralization also applies in general to fibrils, such as to type I collagen (Figure 21d) [219,220].

**Figure 21 pharmaceutics-15-00524-f021:**
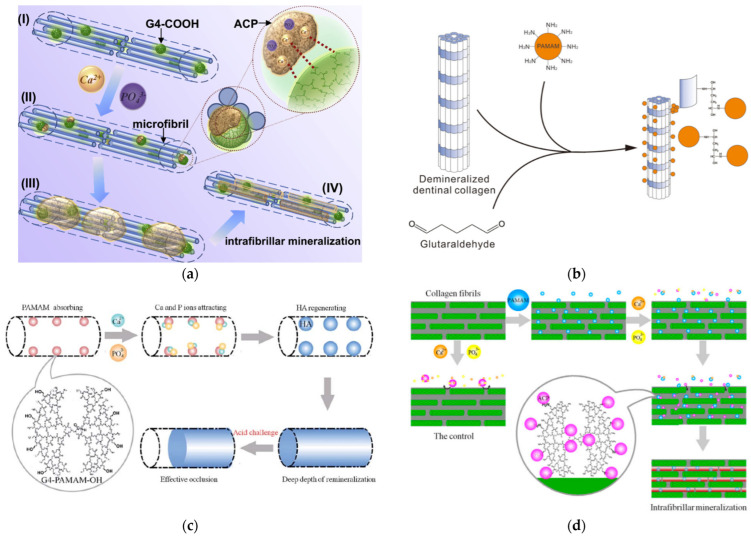
(**a**) Schematic illustration of the bioinspired intrafibrillar mineralization process induced by G4-COOH as artificial proteins. (**I**) G4 PAMAM-COOH binds to collagen fibrils at specific sites of the collagen assembly and retains in it relying on electrostatic force and size exclusion effect. (**II**) G4 PAMAM-COOH captures and stabilizes the metastable amorphous calcium phosphate (ACP) nano-precursors by its carboxyl groups in the mineralization medium. (**III**) G4 PAMAM-COOH acts as the template to make ACP nano-precursors arrange in an ordered manner, resulting in the periodicity of the mineralized fibrils. (**IV**) Small mesocrystals are guided to assembly into larger ones by the immobilized G4 PAMAM-COOH and transformed into apatite nanocrystals, thus the intrafibrillar mineralization along the microfibrils within the collagen matrix is achieved Reproduced with permission from [210]. Copyright: Elsevier Ltd.; (**b**) Reaction scheme that reveals the process of PAMAM dendrimers crosslinking to demineralized dentinal collagen by using glutaraldehyde. Published under Creative Commons license from [213]. (**c**) Schematic demonstration of the effective dentinal tubule occlusion induced by G4-PAMAM-OH. Reproduced with permission from [217]. Copyright: RSC Publishing. (**d**) Schematic demonstration of the mechanism of biomimetic mineralization of collagen fibrils induced by G3-PAMAM. The rectangle represents a collagen fibril. The green cylinders within the rectangle represent microfibrils. The orange, yellow, and purple balls represent Ca^2+^ ions, PO_4_^3−^ ions, and ACP nanoparticles. Under permission from [219]. Copyright: Taylor & Francis.

There are many variants of the above standard pattern, such as pretreatment of dentin samples with Ca(OH)_2_ solution [221], or using amorphous Ca_3_(PO_4_)_2_ nanoparticles [222]. The latter, in combination with G3 PAMAM-NH_2_ (Figure 22) [223], or G3 PAMAM-COOH [224] in an artificial saliva-lactic acid solution, release Ca^2+^ and PO_4_^3−^ ions that strengthen the hardness of the restored dentin to the level of the healthy tissue. An extra beneficial effect of Ca_3_(PO_4_)_2_ nanoparticles is that they may effectively neutralize an acidic environment (pH 4). Incorporated in a dental adhesive comprising pyromellitic glycerol dimethacrylate, 2-hydroxyethyl methacrylate, and ethoxylated bisphenol A dimethacrylate, they may be employed to cure conditions that involve an acidic oral environment, for instance, dry mouth [225,226]. The composition of the adhesive may be upgraded by inorganic fillers such as barium-boro aluminosilicate glass particles to yield templates with recharging capability (Figure 23a) and superb nucleation properties [227,228,229]. Other formulations for Ca_3_(PO_4_)_2_ Nps have also been developed, such as a protein-repellent “bioactive multifunctional composite” comprising 2-methacryloyloxyethyl phosphorylcholine, dimethylamino hexadecyl methacrylate and silver nanoparticles for antimicrobial protection [230].

In a slightly different strategy, the phosphate functionalities were directly bound to the dendritic polymer in an attempt to imitate the role of phosphophoryn. This protein and dentin sialoprotein represent the two most abundant non-collagenous proteins of the dentin matrix [231]. G3 or G4 PAMAM-PO_3_H_2_ were initially chosen because they present similar topological architecture and size. Remineralization of intrafibrillar and interfibrillar reconstituted type I collagen [232] and demineralized human dentin was established in artificial saliva or amorphous calcium phosphate stabilizing, polyacrylic acid (PAA) solution (Figure 23b) [233] and in vivo in the oral cavity of rats [234] To more effectively simulate the operation of non-collagenous dentine matrix protein, phosphorylated G4 PAMAM was combined with the respective carboxylated counterpart. The mechanical properties of hydroxyapatite produced by this blend during the mineralization process of collagen fibrils of natural dentin [235] and recombinant type I collagen fibrils [236] were further improved, resembling at the nanoscale level those of the natural tissues.

The employment of inorganic substrates functionalized with dendritic polymers is equally applied in dentin remineralization. In a reversal of the above-discussed concepts, the occlusion of the dental tubules was effectuated by nano-hydroxyapatite decorated with carboxylated PAMAM (Figure 23c). The binding with collagen fibers is secured by the same mechanism, i.e., via crosslinking with the dentinal fibers (Figure 23d) [237]. An identical beneficiary effect was observed in mesoporous bioactive Ca(NO_3_)_2_ doped glass nanoparticles coated with G4 PAMAM. Moreover, the organic layer did not hinder the release of calcium cations with the soaking in simulated body fluid [238].

Regardless of the remineralization mechanism, a common practice is the inclusion of drugs or other active ingredients into the cavities of the dendritic polymers. Loading of G4 PAMAM-COOH with triclosan antibiotic was proposed for synchronous and protracted antimicrobial protection of the damaged dentin substrate (Figure 24a) [239]. Phosphorylated G3 and G4 PAMAM proved able to solubilize hydrophobic antibacterial apigenin in its cavities to address dental caries. This was attained by preventing the erosion of dentine caused by *Streptococcus mutans* and simultaneous reparation (Figure 24b) [240]. In another example, chlorhexidine was combined with G4-PAMAM-COOH to lessen its cytotoxicity. The mixture was applied as a first coating. The dental adhesive formulation of the second layer was amorphous calcium phosphate nanoparticle adhesive fillers stabilized poly aspartic acid [241]. It was additionally revealed that chlorhexidine reversed the undesired excitatory effect of G4-PAMAM-COOH to matrix metalloproteinase and inhibited its activity [242]. Besides drugs, G3 PAMAM was applied to mechanically exposed pulp teeth to act as a host substrate for pulpine. Clinical trials in 12 patients lasting one and a half years revealed that this gradual layering procedure, apart from more effective dentin mineralization, also enhanced the restoration of the injured pulp tissue [243].

**Figure 22 pharmaceutics-15-00524-f022:**
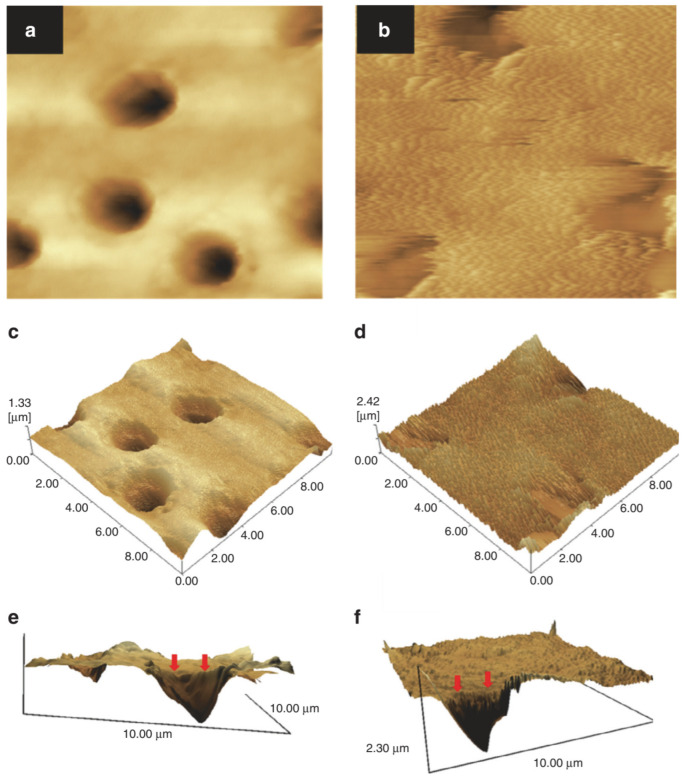
Representative AFM images before and after being immersed in artificial saliva for 4 weeks. AFM images of the demineralized dentin surfaces without treatment (**a**,**c**,**e**) and with the treatment of PAMAM-NH_2_ (**b**,**d**,**f**) after being immersed in artificial saliva for 4 weeks. (**a**,**b**) are the topographical images. (**c**,**d**) are three-dimensional images. (**e**,**f**) are reconstructed from (**a**,**b**) by the software SPM-9700, and a horizontal plane of the remineralization surface and longitudinal profile of the dentinal tubule were observed. The control showed no mineral regeneration. In contrast, PAMAM–NH_2_ induced large amounts of needle-like minerals in dentin. Reproduced under permission from [222]; adapted from [212]. Copyright: WILEY-VCH Verlag GmbH & Co. KGaA, Weinheim.

**Figure 23 pharmaceutics-15-00524-f023:**
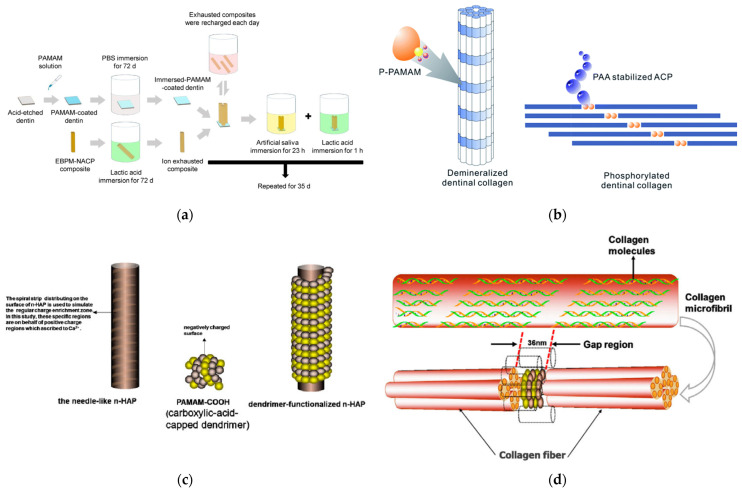
(**a**) Schematic illustration showing how the PAMAM -adhesive composite (EBPM)- Ca_3_(PO_4_)_2_ nanoparticles (NACP) composite method was used to induce long-term dentin remineralization. Reproduced with permission from [229] Copyright: The Academy of Dental Materials. (**b**) Schematic demonstration of the effective biomimetic remineralization of demineralized dentinal collagen fibrils using G3-PAMAM-PO_3_H_2_ dendrimers and polyacrylic acid (PAA). Under permission from [233] Copyright: RSC Publishing. (**c**) Schematic illustrations of the n-HAP surface functionalized by Carboxylic acid-capped dendrimer (PAMAM-COOH). The spiral strip region on the surface of nHAP is used to imitate the positive charge enrichment zone. Based on the electrostatic forces, the crystal surface is regularly coated with the dendrimer after the addition of PAMAM-COOH. (**d**) Mechanism of PAMAM dendrimers (PAMAM-COOH) crosslinked to demineralized dentinal collagen in certain areas—gap region. Five collagen molecules form a collagen microfibril by self-assembly, and the bundling of several microfibrils forms a collagen fibril. By electrostatic force and size exclusion effect, the dendrimer-functionalized n-HAP can attach to the gap region which is between two consecutive collagen molecules, and be stabilized. The dashed part indicates mineral crystals portions (n-HAP). As limited by the dimensions and shapes, mineral crystal portions (n-HAP) cannot infiltrate into the collagen fiber but spread along the surface of the collagen fibrils. Both were reproduced under permission from [237]. Rights managed by Taylor & Francis.

**Figure 24 pharmaceutics-15-00524-f024:**
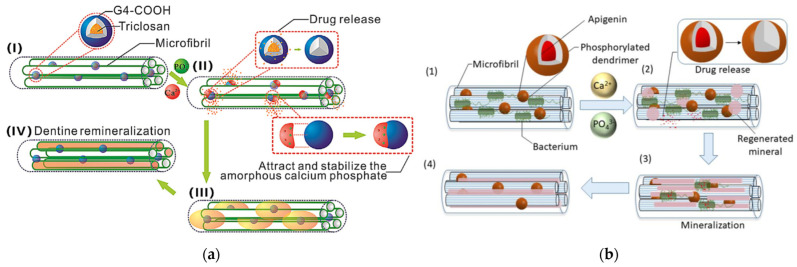
(**a**) Schematic illustration of the sustained release of encapsulated triclosan from the formulation and the simultaneous mineralization induced by G4 PAMAM-COOH. (**I**) The triclosan-loaded G4 PAMAM-COOH is adsorbed on the dentine microfibrils. (**II**) Encapsulated triclosan is released during the incubation in an artificial saliva solution. Meanwhile, the anionic carboxylic groups of G4 PAMAM-COOH attract and stabilize the amorphous calcium phosphate (ACP) in an artificial saliva solution. (**III**) The mineralized fibrils could emerge due to the template role of dendrimers. (**IV**) The mineralized fibrils become larger, making ACP arranged in an ordered manner and resulting in HA crystallization Reproduced with permission from [239]. Copyright: Elsevier B.V. (**b**) Phosphoryl-terminated PAMAM dendrimers loaded with apigenin induce dentine tubules occlusion through mineralization, and released apigenin prevents further erosion of dentine by bacteria. Reproduced with permission from [240]. Copyright: Elsevier B.V.

### 5.3. Enamel

Enamel, the other major hard tissue of the tooth, was submitted to treatment approaches similar to dentin with analogous successful results [244]. An investigation for both components was conducted synchronously and side by side, beginning with anionic carboxylated PAMAM. The third [245] and the fourth generation [246] counterparts induced the crystallization of rod-like hydroxyapatite crystals on the etched enamel surface in the same orientation as the long axis of enamel crystals. In contrast to dentin, the different PAMAM terminal groups produced a differentiation in the enamel lesion remineralization percentage: (PAMAM-NH_2_ 76.42 ± 3.32%), (PAMAM-COOH 60.07 ± 5.92%), and (PAMAM-OH 54.52 ± 7.81%) was quantified in bovine enamel specimens in a simulated oral environment [247]. The enamel surfaces, remineralized by the fifth generation of the two best-performing organic templates (PAMAM-NH_2_, PAMAM-COOH), were subsequently tested against adhesion and biofilm formation from *Streptococcus mutans*. Both dendrimers resisted bacterial attacks, highlighting their potential in preventing secondary caries [248].

The phosphorylated PAMAM option was implemented on human tooth enamel. The G4 counterpart possesses the same peripheral phosphate groups and size as those of amelogenin, the most important protein in the natural process of enamel. For this reason, G4 PAMAM-PO_3_H_2_ is tightly adsorbed on the enamel substrate and the bioinspired remineralization process in artificial saliva or the oral cavity of rats proceeds more efficiently in comparison to PAMAM-COOH, with a crystalline hydroxyapatite layer thickness of about 11.23 μm instead of 6.02 μm (Figure 25A) [249]. Another amphiphilic PAMAM dendron bonded with stearic acid at the focal point and coupled peripherally with aspartic acid moieties forms spherical organizations in solution. These undergo further aggregation as a function of concentration to linear chains such as amelogenin (Figure 25B). In this way, HAP nucleation may follow desired orientations similar to those encountered in enamel [250]. Even a simple carboxy G4.0 PAMAM-COOH derivative may present a microribbon hierarchical organization similar to the amelogenin prototype, with a suitable ion chelating cation, such as in an aqueous ferric chloride solution (Figure 25C) [251]. A similar decoration of G3.5 PAMAM-COOH with alendronate groups (Figure 26a) bearing two phosphate functionalities was made to enhance the adsorption of the dendrimer to the enamel layer [252].

In a contiguous attempt to imitate the role of another protein (salivary statherin), G4 PAMAM was modified by the N-15 peptide. Then, it was incorporated into formulations containing calcium phosphate nanoparticles and the same adhesive resin used as described above [225] for the remineralization of dentin, with equally successful performance [253]. An enamel-specific water-insoluble antibacterial and antibiofilm agent, honokiol, may also be included in PAMAM-COOH and then released in a controlled profile. Anticaries’ activity was established through planktonic growth assays and in vivo in male Sprague Dawley rats (Figure 26b) [254]. The research on the potential of dendritic polymers for use in dentistry is summarized in Table 4.

**Figure 25 pharmaceutics-15-00524-f025:**
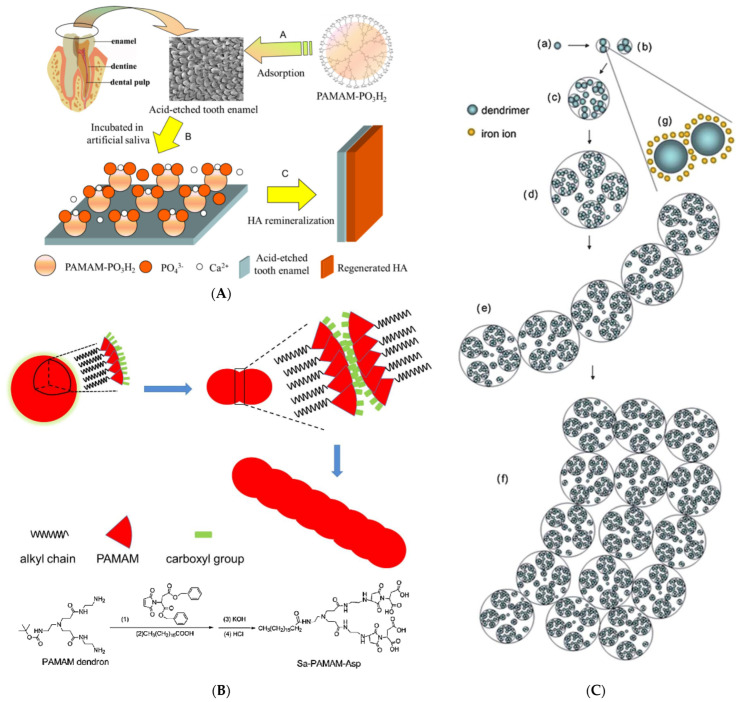
(**A**) Schematic demonstration of the adsorption of PAMAM-PO_3_H_2_ on the surface of tooth enamel and the subsequent in situ remineralization of hydroxyapatite. Reproduced under permission from [249]. (**B**) The modification of PAMAM dendron with aspartic acid-stearic acid and schematic illustration of the linear chain aggregation of the nanospheres. Reproduced with permission from [250]. Copyright: ROYAL SOCIETY OF CHEMISTRY (**C**) Schematic illustration of the self-assembly process of G4.0-COOH PAMAM. (**a**) The single G4.0-COOH PAMAM dendrimer molecule is a globular unit. *R_h_* = 4.4 nm. (**b**,**c**) Oligomers of G4.0-COOH PAMAM dendrimers. *R_h_* = 6–10 nm and 40–80 nm for different stages. (**d**) Nanosphere structures are formed through the association of oligomers and monomers. *R_h_* = 160–200 nm. (**e**) Further association of nanospheres results in a larger assembly to form nanosphere chains. (**f**) The organization of the nanosphere chain forms a microribbon structure. (**g**) Interconnection of G4.0-COOH PAMAM dendrimers by iron ion chelation. Reproduced with permission from [251]. Copyright: Royal Society of Chemistry.

**Figure 26 pharmaceutics-15-00524-f026:**
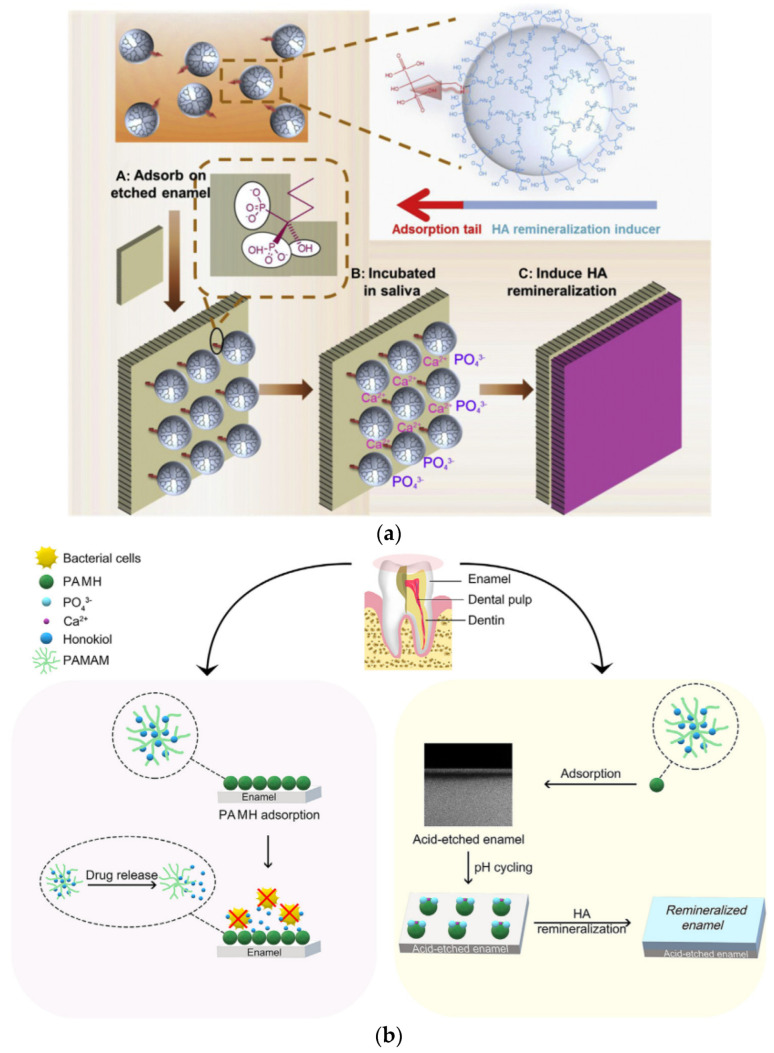
(**a**) Schematic demonstration of the specific adsorption of alendronate-PAMAM-COOH on the surface of tooth enamel and the subsequent in situ remineralization of HA. Reproduced under permission from [252]. Copyright: Elsevier Ltd. (**b**) Schematic demonstration of the mechanisms of PAMAM complex with honokiol (PAMH) for antibacterial and remineralizing effects on enamel, both of which account for anticaries management in the animal experiments of the current study. Reproduced with permission from [254]. Copyright: The Academy of Dental Materials.

## 6. Treatment of Eye Related Conditions

### 6.1. Remediation of Cornea Incisions and Injuries

In parallel to the endeavors made for the synthesis of artificial biomolecules, research has also begun in the field of soft tissue (re)connecting. Post-surgical corneal lacerations provided the ideal experimental substrate and the first proposed solution related to linear poly(ethylene glycol) core-dendritic polyesters (Figure 27a). External hydroxyl groups were replaced by photopolymerizable methacrylate groups [255] and the resulting cross-linked gels exhibited adequate adhesive properties to the relevant tissue to seal corneal incisions and replace conventional sutures [256]. The effectiveness of the sealant was found to be dependent on dendrimer concentration and the Mw of the linear core, with the intermediate values (Mw 10,000 and 20% wt/vol) being more efficient [257]. Specimens from this self-photo-cross-linkable polymer family were tested in vivo in white leghorn chickens. After their application to corneal wounds inflicted by 4.1-mm keratome and activation by argon ion laser, they repaired the injured tissue in one minute, five times faster than conventional sutures, eliminating moreover completely corneal scarring [258].

In an alternative chemoselective crosslinking, lysine-based peptide dendrons with four or eight terminal cysteine residues reacted with poly(ethylene glycol dialdehyde) [259] poly(ethylene glycol) diester-aldehyde [260], poly(ethylene glycol)-butyric dialdehyde [261,262] propanoic dialdehyde, or 2-oxoethyl succinate-functionalized poly(ethylene glycol) [262]. An initial thiazolidine bond between the aldehyde and aminothiol groups was transformed into pseudoproline linkage via an O,N-acyl migration (Figure 27b) and afforded the respective hydrogels. The recommended application was the postoperative treatment of cataract incisions. Both variants were tested in human eyes against laser in situ keratomileusis flaps and both proved successful in securing them [263]. The most advanced conception consisted of a two-layer design. The upper lamella contains type I collagen, chemically connected to methacrylic acid and 2-hydroxyethyl methacrylate hydrogel copolymer for the immobilization and proliferation of corneal epithelial cells. This is superimposed on a hydrogel formed by the classic hybrid linear poly(ethylene glycol) core-dendritic polyesters modified by methacrylate groups and PEG (3400) acrylamide with cysteine-arginine-glycine-aspartic acid tetrapeptide for the propagation of corneal fibroblasts and the integration with the underlying stromal cells. The resulting implant is suggested as an inlay for lamellar keratoplasty [264].

**Figure 27 pharmaceutics-15-00524-f027:**
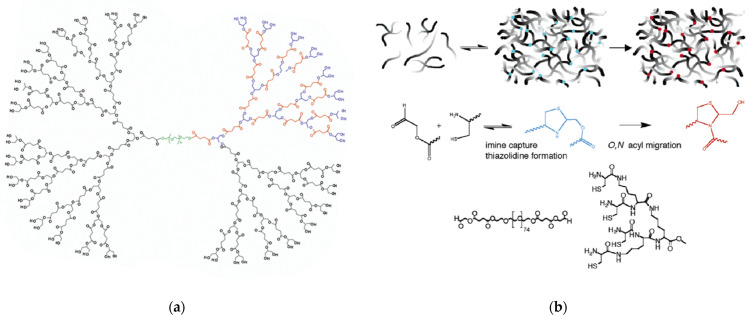
(**a**) Polyester dendrimer with a linear poly(ethylene glycol) core. Reproduced with permission from [255]. Copyright: American Chemical Society; (**b**) Crosslinking mechanism for the pseudoproline ligation between poly(ethylene glycol diester-aldehyde and N-terminal cysteine lysine dendrons for hydrogel formation. Reproduced with permission from [260] Copyright: American Chemical Society.

### 6.2. Corneal Tissue (re)Creation

The construction of scaffolds for tissue engineering is one of the first applications of artificial biomolecules and, more specifically, G2 Poly(propylene imine) PPI dendrimer crosslinked collagen discussed in Section 2.1 [124]. These matrices proved non-toxic in contrast to those built with the glutaraldehyde alternative crosslinker. On top of that, they are promising materials for the adhesion and proliferation of human corneal epithelial cells to obtain a synthetic cornea [265]. The same material exhibited high heparin retention. When loaded with basic fibroblast growth factor (FGF-2), the release rate in the first 2 weeks was 20–60% depending on the cross-linking density [266]. Corneal stromal cell stimulation may be readily achieved by fine-tuning this delivery to heal wounds or produce corneal tissue. Replacement of FGF-2 with heparin-binding epidermal growth factor (HB-EGF) increased its release rate up to 90% and the proliferation of human corneal epithelial cells (HCEC) [267]. Functionalization of the PPI dendrimer by the laminin cell adhesion peptide YIGSR improved corneal epithelial cell adhesion and growth. Furthermore, it increased the number and the length of neurites that extended from dorsal root ganglia cells [268].

Before closing this session, it is necessary to mention a very recent application that marries dendrimers mimicking elastin conformation with neural crest cell differentiation to corneal endothelial cells (CEC). Thermo-responsive G4 PAMAM was synthesized by reaction of the terminal amino groups with the NHS functionalities of a PEG (1000) chain. Then, attachment of the active maleimide group of the second PEG end through a cysteine residue to the RGD tripeptide was performed to improve cell adhesion. Finally, a tetramer of the standard sequence Val-Pro-Gly-Val-Gly was added for elastin mimicry (Figure 28) [269]. This bioinspired substrate proved practical for homogenous spreading and fast proliferation of the cells. In the first stage, human adipose-derived mesenchymal stem cells (hADSCs) were converted to neural crest cells, and then on the surface of the dendritic derivative, differentiation to CEC occurred. Isolated sheets from this material are expected to alleviate problems arising from CEC’s limited in vivo division and cure various conditions, such as corneal dystrophies and glaucoma. The propositions for dendritic polymers related to the treatment of eye injuries are summarized in Table 5.

## 7. Skin

### 7.1. Wound Treatment by Hydrogels

Skin injury healing comprises procedures that are very similar to cornea restoration and thus involve analogous treatments. For instance, coupling poly(L-lysine) dendrigrafts (DGL) with linear PEG, bearing cross-linking functionalities yield hydrogels that increase human dermal fibroblast adhesion as a function of the dendritic polymer ratio. Furthermore, porosity may be induced by paraffin microsphere porogen and subcutaneous implantation in mice causes cell infiltration and blood vessels invasion into the pores [270]. A cross-linked hydrogel is formed based on the thiol–thioester reaction. This is a dynamic equilibrium that reversibly forms and breaks covalent bonds, between a dendron with terminal thiols attached by PEG spacers and poly(ethylene glycol disuccinimidyl valerate [271]. Adhesion to the skin and mechanical properties of the product were very promising as well as its elasticity and hydrophilicity. Due to the temporary character of the gel bond network, it may also be easily removed by washing it with a thiolate solution. Reversible crosslinking was also achieved by a simple dynamic thiol–aldehyde reaction. A hyperbranched thiol-terminated polymer (HBPTE) was directly mixed with polyethylene glycol bearing benzaldehyde moieties at each end (PEGCHO) (Figure 29). The resulting gel could undergo a thermal transition to the liquid state at 50 °C which facilitated its spreading throughout the wound. For this reason, it readily bound in vivo to mice skin tissues and accelerated their restitution [272]. Through thiol–ene coupling, a diversification of the above method, a dendritic–linear–dendritic copolymer from linear PEG and G3-G6 hyperbranched 2,2-bis(hydroxymethyl) propionic acid with terminal allyl groups was cross-linked with tris [2-(3-mercapto propionyl oxy)ethyl] isocyanurate with the aid of high-energy visible light (Figure 30a). The soft tissue adhesive patches fabricated by the addition of surgical mesh exhibited mechanical integrity, and satisfactory adhesion to porcine skin and permitted the proliferation of mouse embryonic fibroblasts. Besides skin healing, the above properties render these materials promising solutions for internal wound repairs [273].

Alternatively, by reaction with thioglycolic acid, the thiolate groups may be incorporated into the PEG moiety (Figure 30b). This modification provided faster dissolution (Figure 30c) and was proposed for dressing burn wounds and easy removal without additional burdens [274]. An analogous hydrogel with amino groups at the place of thiols and thus permanent amide groups exhibited an equally adequate performance when tested against commercially available sealants [275]. In an uncustomary skin application, biological bandages with progenitor skin cells and growth factors were combined with peptide (arginine-lysine) polycationic dendrimers. These macromolecules exhibit anti-microbial activity against *Pseudomonas aeruginosa* and unexpected angiogenic activity and were used for the treatment of deep burn wounds [276].

### 7.2. Dendritic Polymers as Cell Nanocarriers to the Skin

G5 PAMAM dendrimers functionalized with adhesion molecules sE-selectin (sE-Sel) and Vascular Endothelial Growth Factor (VEGF) (Figure 31a) may also act as nanocarriers for appropriate cells possessing wound healing properties such as bone marrow cells. These functionalities are recognized by complementary receptors expressed from the endothelium of injured tissue cells: E-selectin ligand CD44 and VEGF receptor, respectively (Figure 31b,c). Once brought into the vicinity of the wound, they reach their intended tissues through transendothelial migration and extravasation and accomplish their therapeutic task (Figure 31d). Apart from cutaneous damages, this technique may also be employed in corneas as well as any other similar therapeutic implementation [277]. 

### 7.3. Scaffolds for Skin Tissue Regeneration

Scaffolds present a useful alternative for skin tissue engineering. When G4 PAMAM was combined with gelatin it enhanced its swelling and biodegradation properties. Adhesion and proliferation of both collagen type I fibroblasts and keratinocytes was boosted [278]. The surface roughness and hydrophilicity of poly(L-lactic acid) (PLLA) films and electrospun scaffolds were substantially increased by the embodiment of G2, G3, and G5 PAMAM and PPI dendrimers, and their biodegradation was protracted. Human dermal fibroblast cells cultivated in the modified surfaces demonstrated higher attachment properties and homogeneous spread morphology (Figure 32) [279].

On the other hand, PAMAM dendron decorated nanoparticles (Nps) govern self-assembly in collagen analogs. For this reason, metal oxide Nps (12–25 nm), when encircled by the third-generation counterpart via the standard aminopropyl triethoxy silane linker, increase the physicochemical mechanical properties of the resulting collagen matrices (Figure 33a) as well as cell viability. Faster re-epithelization and superior wound healing properties with no scars were observed in vivo in Wistar albino rats in the order ZnO > TiO_2_ > CeO_2_ > SiO_2_ > Fe_3_O_4_ (Figure 33b) [280]. The influence of the nanoparticle shape was investigated with ZnO and respective G1 dendrons (Figure 33c), highlighting spherical Nps as the best (Figure 33d) [281]. The same idea works for carbon-based nanomaterials: spherical fullerenes (0D), tubular carbon nanotubes (CNT, 1D), planar graphene oxide (2D), and reduced graphene (2D) (Figure 33e). The one-dimensional nanotubes exhibit the optimal therapeutic action due to their orientation along the collagen’s linear axis (Figure 33f) [282]. Skin therapies involving dendritic polymers are summarized in Table 6.

## 8. Articular Cartilage Tissue Reparation

Chronologically, cartilage is the second soft tissue, next to the eye, that has received extensive interest from regenerative medicine scientists. This is not only due to the frequency of injuries or conditions such as osteoarthritis but also because of the particular nature of the relevant cells. Hydrogel scaffolds made from photo cross-linkable dendritic polymers were once more the first thought. A typical dendric polymer with PEG core and methacrylated poly(glycerol succinic acid) peripheral groups, that is commonly used for corneal tissue reparation [283] proved ideal for filling the irregular gaps of the injured tissue and for contributing to chondrocytes proliferation and neo-cartilaginous tissue chondrogenesis [284]. Photopolymerizable dendrimers with a similar core; branches made from biocompatible succinic acid, glycerol, and β-alanine and carbamate group terminals, present a further option for hydrogel matrices. They are susceptible to cell infiltration and their ex vivo application to the osteochondral defect in New Zealand white rabbits produced substantial quantities of collagen II and glycosaminoglycans that contributed to the repair of hyaline cartilage tissue [285].

The chondrogenic differentiation of stem cells is another strategy for the reconstitution of cartilage tissue. PAMAM partially modified with polyethylene glycol moieties may immobilize kartogenin (KGN) into the dendritic cavities or bond it to the PEG chains. In the second stage, the protein is delivered to the cytoplasm of mesenchymal stem cells to induce their differentiation (Figure 34a). Absorption of kartogenin into the dendrimer proved the most effective method, as established by the intensity of the β nuclear localization chondrogenesis marker [286]. In a combination of these two above-described methods, hydrogels based on gelatin methacrylate enveloped by PAMAM with methacrylic anhydride end groups (PAMAM-MA) supported adipose-derived stromal stem cells. Subsequently, they underwent osteogenic, adipogenic, and chondrogenic differentiation, expressing related genes and proteins promoting in vivo articular cartilage regeneration in 6-week-old female Sprague Dawley rats (Figure 34b) [287]. In conclusion, in a targeted cell delivery example, G4 PAMAM coating to biodegradable polyester poly(lactide-co-glycolide) microspheres proved beneficial for the culture of sheep articular cartilage chondrocytes. The resulting composite mixed with collagen type I gels allowed cell transplantation to damaged cartilages of nude mice and hyaline tissue reconstitution without modulation of their phenotype [288]. The dendritic polymer substrates discussed as appropriate for cartilage cells are summarized in Table 7.

## 9. Hepatic Aortic Neural and Pancreatic Tissue Engineering

The organization of hepatic cells is the most crucial factor for the assembly of a biological artificial liver system. The three attempts to aggregate hepatic cells without a solid matrix analyzed in the “Dendritic Glues for Cell Aggregation” subsection constitute the background research of the initial approaches in this field [145,147,148]. Scaffold technology is once more applied to liver tissue development. Electroactive nanofiber matrices constructed from polycaprolactone and aliphatic polyester/polypyrrole G4, G6 hyperbranched polymers may closely imitate the environment of the extracellular matrix and enhance the proliferation and differentiation of HEP G2 cells (Figure 35a) [289]. Similar fibrous templates were prepared from poly(glycolic acid), which is a fitting substrate for the adhesion of nerve cells. The fibers were covered by a polylysine dendrigraft that supported hippocampal neuron development more efficiently than linear polylysine (Figure 35b) [290].

Hepatic injuries, together with aortic, corneal, cartilage, and skin are all in the same class of conditions that may be treated by the application of suitable hydrogel sealant. The thiol-thioester reversible model that permits the rapid redissolution of the gel and the re-exposure of the trauma is also the basis for this adhesive. For the cross-linking reaction, a lysine dendron with thiol end groups reacts with maleimide-terminated PEG (Figure 35c). The resulting gel was able to limit blood loss by 33% in rats with induced hepatic hemorrhage, whereas the corresponding percentage for aortic hemorrhage was 22% [291].

As was mentioned in the study on the treatment of eye-related conditions for corneal fibroblasts, the modification of PAMAM with the suitable peptide is beneficial for their proliferation. A modified oligopeptide (HAS-28) derived from neuroligin-2 (Figure 35d) attached to G5 PAMAM proved very advantageous for pancreatic β-cell culture, functional maturation, as well as glucose-stimulated insulin secretion [292]. Another peptide sequence, fibroblast inhibitor (isoleucine-lysine-valine-alanine-valine (IKVAV)) was attached peripherally to a polylysine dendrimer. The complex was assimilated to pre-crosslinked collagen films that selectively favored the adhesion and proliferation of rat Schwann cells, and obstructed human dermal fibroblast development. In contrast, non-cross-linked collagen films or counterparts crosslinked after the incorporation of the dendrimer did not show selectivity. Possible applications reside in the treatment of larger peripheral nerve injuries while preventing detrimental fibrotic scarring [293]. The same Ile-Lys-Val-Ala-Val (IKVAV) oligopeptide together with Arg-Gly-Asp (RGD) and Tyr-Ile-Gly-Ser-Arg (YIGSR) when attached to G5 PAMAM achieved neurite differentiation of pheochromocytoma (PC12) cells and increased their proliferation [186].

Closing this review article, the authors would like to highlight the following very innovative artificial liver application. Simple G4 PAMAM grafted to ordinary cellulose filter paper by crosslinking with glutaraldehyde permits high adhesion, proliferation, and viability of HepG2 cells. This observation permitted the manufacturing of a 3D liver model for drug screening and liver-associated diagnostics. Proof of concept has been performed against hepatotoxins [294]. The interactions of the dendritic polymers with hepatic aortic neural and Pancreatic cells are summarized in Table 7.

**Figure 35 pharmaceutics-15-00524-f035:**
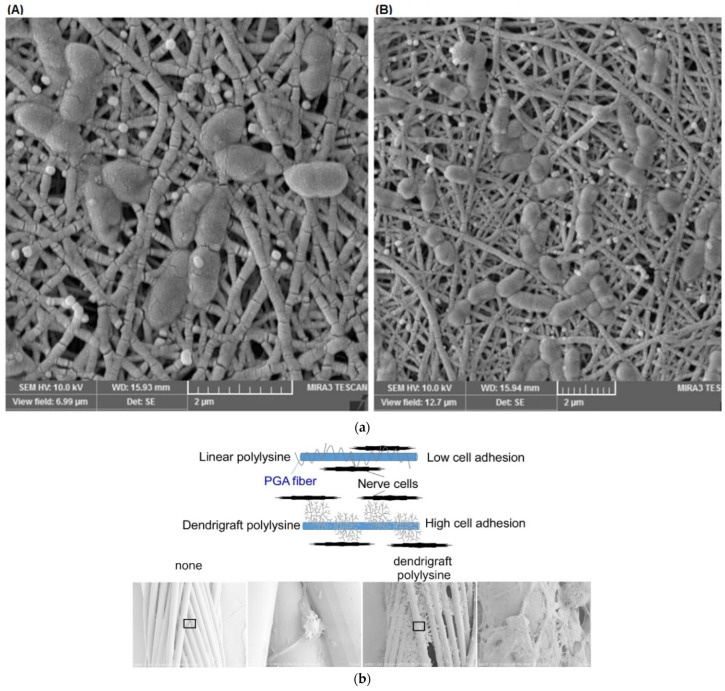
(**a**) SEM images of human HEP G2 cells on HAPG4PPy/PCL (**A**) and HAPG6PPy/PCL (**B**) electrospun nanofibers. Reproduced with permission from [289]. Copyright: Taylor & Francis; (**b**) Schematic comparison of hippocampal neuron adhesion to (**A**) linear and (**B**) dendrigraft polylysine-coated poly(glycolic acid) fibers and SEM images of primary cultured hippocampal neurons on fibers with and without coating with dendrigraft polylysine (G5) Expanded images in the left panels are shown in the right panels. Reproduced with permission from [290]. Copyright: Wiley Periodicals, Inc. (**c**) Chemical reaction scheme for the hydrogel formation upon mixing the lysine dendron with the PEG crosslinker, and for the subsequent dissolution with a cysteine methyl ester. Reproduced with permission from [291]. Copyright: Elsevier Inc. (**d**) Chemical structure of modified HSA-28 peptide. Reproduced with permission from [294]. Copyright: Royal Society of Chemistry.

**Table 7 pharmaceutics-15-00524-t007:** Cartilage hepatic aortic neural and pancreatic cell development by the mediation of dendritic polymers.

Dendritic Polymer	Modification	Substrate	Function	Ref.
PAMAM G4	Bovine collagen type I	Poly(lactide-co-glycolide) Microspheres	Chondrocytes Proliferation- Articular Cartilage Tissue Regeneration	[288]
PEG core methacrylated poly(glycerol succinic acid)	PEG core photo–cross-linking	Hydrogel	Chondrocytes Proliferation- Cartilage Tissue Regeneration	[284]
PEG core poly(glycerol succinic acid), poly(glycerol beta-alanine)	Methacrylate ester photo–cross-linking	-	Osteochondral defects repair	[285]
G4 Dendrigraft polylysines	-	Poly(glycolic acid) fibrous scaffolds	Hippocampal neurons (nerve cells)	[290]
G4 PAMAM	PEG	Kartogenin	Chondrogenic differentiation of mesenchymal stem cells	[286]
Thiol-terminated dendron	Maleimide end-capped PEG crosslinker	Hydrogel sealant	Hepatic and aortic trauma	[291]
G4 PAMAM		Filter paper functionalized with glutaraldehyde	Liver HepG2 cells	[294]
G5 PAMAM	Neuroligin-2-derived peptide		Pancreatic β-cells’ proliferation	[292]
Hyperbranched polyester	Polypyrrole end groups polycaprolactone	Nanofiber Scaffold	HepG2 cells Liver	[289]
Poly ε-lysine dendrimers	Ile-lys-val-ala-val	Pre-crosslinked collagen	Rat Schwann cells proliferation, human dermal fibroblasts inhibition, selective neural cell response	[293]
PAMAM	Methacrylic anhydride	Gelatin methacrylate hydrogel	Cartilage defect repair, in vivo cartilage defect repair	[287]

## 10. Conclusions

The vast majority of the research performed in the field of tissue engineering assisted by dendritic polymers concerns dendrimers and, more specifically, PAMAM and its functionalized derivatives. Symmetry, precise molecular weight, and defined structure play a crucial role in the reproducibility of critical biological responses. Yet, in some cases, cheaper hyperbranched polymers may also provide solutions that are equivalent and notably economically feasible. In the last twenty years, enormous progress has been achieved in answering the fundamental question of why so many proteins are involved in a seemingly simple process: the nucleation of oriented and controlled HAP biomineralization in bones, dentin, and enamel. Many efficient restorative products are comparable to or better than those commercially available and are closer to being utilized in the relevant markets.

Mimicry of biomolecules such as collagen and elastin has also benefited from the substantial successes of dendritic polymer research in the functionalization of PPI and PAMAM with the appropriate peptides. Moreover, significant advancements have been performed in the wound healing of many soft tissues, both external for instance cornea or skin, and internal, such as for articular cartilage, hepatic, aortic, neural, and pancreatic. This is achieved by hydrogels consisting usually of a mixture of linear and dendritic polymers bearing amino acid sequences. A multitude of crosslinking strategies has been developed, including reversible variants such as the thiol thioester mechanism.

However, we are a long way from reproducing the complexity of hierarchical structures in bone and teeth and even further from approaching the level of organized tissues and entire organs. Promising hybrid organic-inorganic scaffold compositions for cell differentiation and their structural arrangement open an optimistic perspective for the improvement of the efficacity of dendritic polymers in tissue regeneration.

## Figures and Tables

**Figure 1 pharmaceutics-15-00524-f001:**
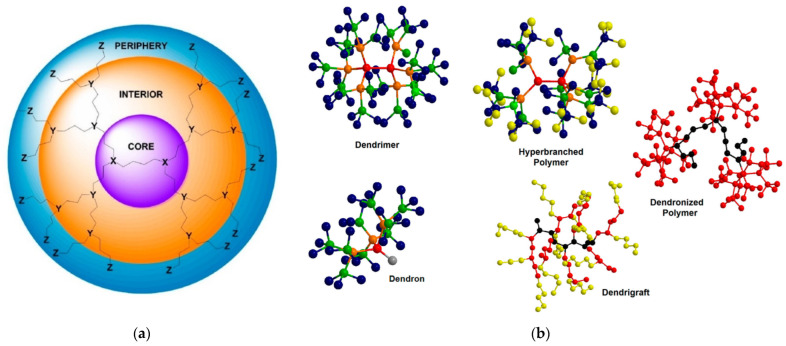
(**a**) Schematic representation of the three structural parts of a dendritic polymer. Schematic representation of the three structural parts of a dendritic polymer. Reproduced under creative commons license from [31]. (**b**) Schematic representation of the categories of dendritic polymers. Each sphere represents a monomer. Different colors depict the different polymerization generations. Reproduced with permission from [32]. Copyright: Taylor & Francis.

**Figure 2 pharmaceutics-15-00524-f002:**
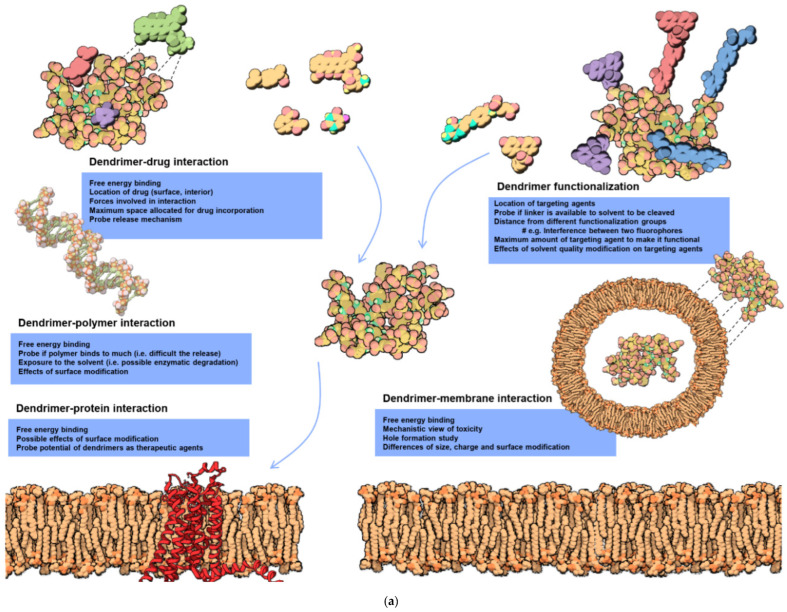
(**a**) Biomedical applications where functionalized dendritic polymers are important to probe biological interactions. Reproduced under creative commons license from [66]. (**b**) A dimensionally scaled comparison of a series of poly(amidoamine) (PAMAM) dendrimers (G = 4–7) with a variety of proteins, a typical lipid-bilayer membrane, and DNA, indicating the closely matched size and contours of important proteins and bio-assemblies. Reproduced with permission from [67]. Copyright: Elsevier Science Ltd.

**Figure 3 pharmaceutics-15-00524-f003:**
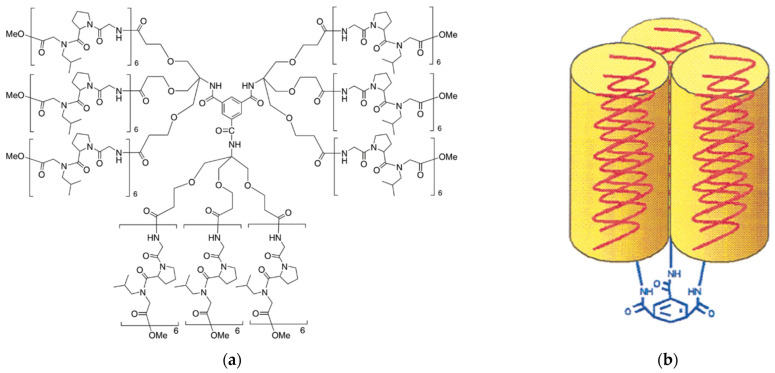
(**a**) The collagen mimetic dendrimer TMA[TRIS[(Gly-Pro-Nleu)_6_-OMe]_3_]_3_. (**b**) Triple helices conformation of the amino acid clustering emanating from the trimesic acid core, both reproduced with permission from [119]. Copyright: American Chemical Society.

**Figure 4 pharmaceutics-15-00524-f004:**
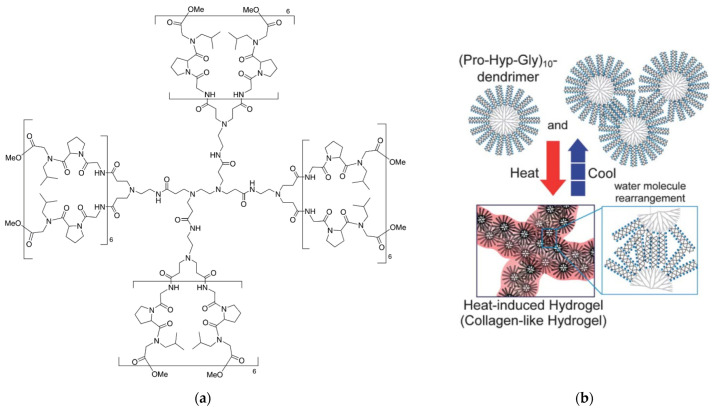
(**a**) Structure of PAMAMG0.5[(Gly-Pro-Nleu)_6_-OMe]_8_. Reproduced with permission from [120]. Copyright: Elsevier Ltd. (**b**) Gelation mechanism of the temperature-dependent G4 PAMAM [(Gly-Pro-Hyp)]_10_ Reproduced with permission from [123]. Copyright: Royal Society of Chemistry.

**Figure 5 pharmaceutics-15-00524-f005:**
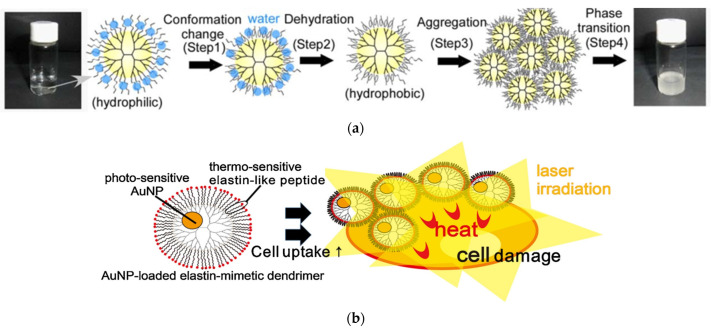
(**a**) Possible mechanism of phase transition in elastin mimetic dendrimers. Reproduced with permission from [126]. Copyright: Wiley Periodicals, Inc. (**b**) Dual stimuli: photo and thermo-sensitive dendrimer derivatives. Reproduced with permission from [127]. Copyright: Elsevier Ltd.

**Figure 7 pharmaceutics-15-00524-f007:**
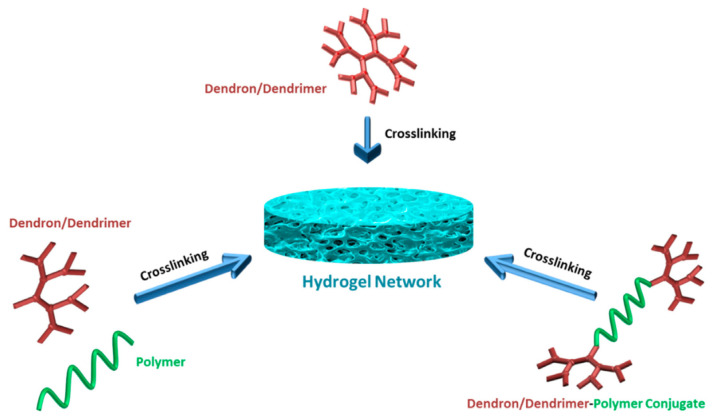
Approaches toward the fabrication of hydrogels using dendron- and dendrimer-based building blocks. Reproduced under creative commons license from [139].

**Figure 8 pharmaceutics-15-00524-f008:**
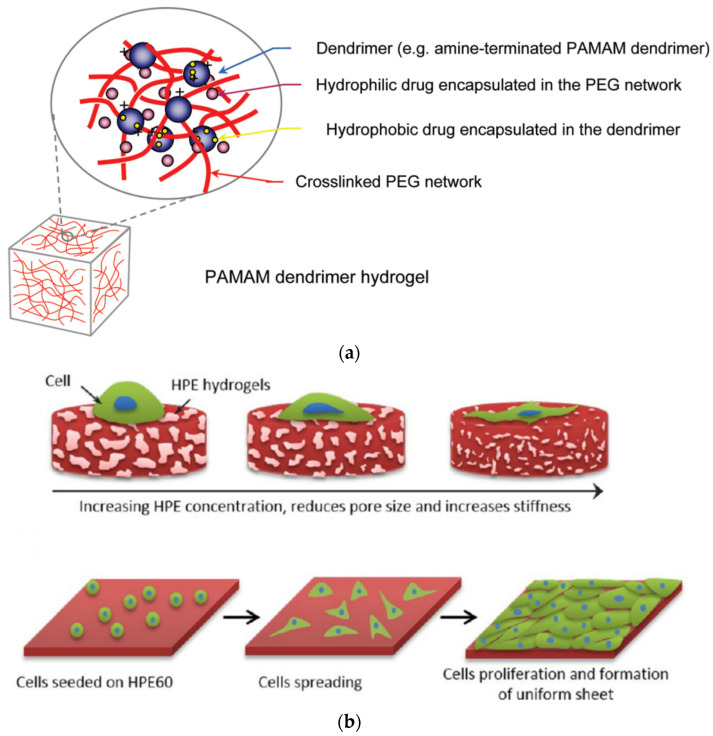
(**a**) Schematic of a cross-linked PAMAM dendrimer network. Reproduced with permission from [140]. Copyright: American Chemical Society. (**b**) Schematics showing the increase in cell adhesion and spreading can be attributed to the higher stiffness of hydrogels resulting from an increase in HPE concentration. The controlled cell adhesion properties of HPE60 can be utilized to fabricate uniform cell layers. Reproduced with permission from [141]. Copyright: American Chemical Society.

**Figure 9 pharmaceutics-15-00524-f009:**
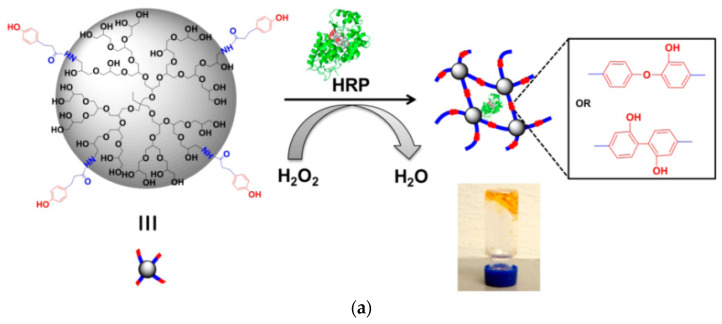
(**a**) Formation of PG Hydrogels by HRP Cross-Linking; (**b**) PG gel formation via gluco-δ-lactone triggered by glucose. Both were reproduced with permission from [142].

**Figure 10 pharmaceutics-15-00524-f010:**
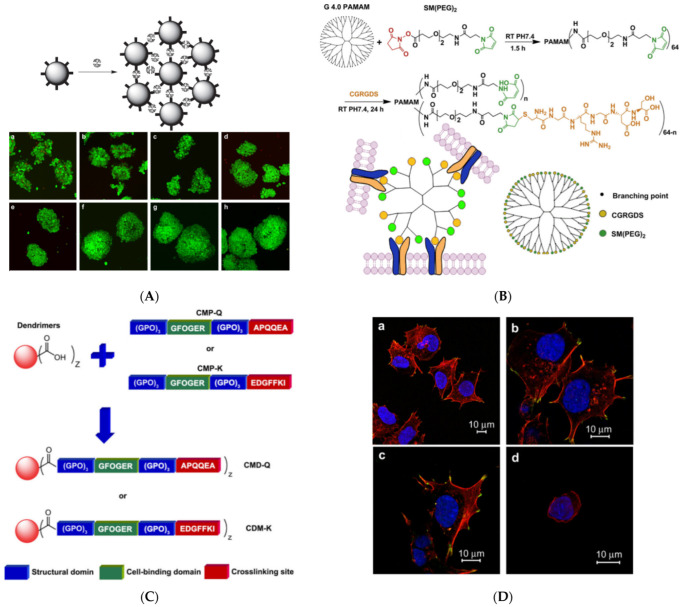
(**A**) Cellular aggregate formation using dendrimer hydrazide as an inter-cellular linker. Surface-modified cells are cross-linked by dendrimer hydrazide to rapidly form the assemblies; introduced aldehyde residues of cell surface modification are schematically indicated by the peripheral short lines; hydrazide dendrimer and confocal images of live (green) and dead (red) HepG2 cells in MCAs on days 0, 1, 2, 3, 4, 5, 6 and 7, respectively, indicated high cell viability, and an increase in MCA size (**a**–**h**) Reproduced with permission from [145] Copyright: Elsevier Ltd. (**B**)A Synthetic scheme for modification of PAMAM with RGD via a two-step reaction using a heterobifunctional coupling reagent, SM(PEG)2, and a model of RGD-PAMAM interacting with different cells and functioning as “molecular glue” at the multicellular interface. Reproduced with permission from [146]. Copyright: Elsevier Ltd. (**C**) Collagen-mimetic peptides (CMP-Q or CMP-K) supplemented with a cell-binding sequence (GFOGER) and the identified EDGFFKI or APQQEA substrate sequence were conjugated onto a PAMAM dendrimer to create a cross-linkable ‘‘biomimetic collagen’’. Z denotes the number of peripheral functional groups of the dendrimers available for tethering peptides covalently nearby thus promoting intermolecular interactions and folding. (**D**) Cytoskeletal organization of Hep3B cells as a function of substrates: (**a**) calf-skin collagen, (**b**) crosslinked collagen-mimetic dendrimers (X-CMD), (**c**) PAMAM G1.5–[(GPO)3GFOGER(GPO)3EDGFFKI]7 (CMD-K), and (**d**) GFOGERGGG (CMP00). Cells were fixed and stained for actin stress fibers (TRITC–phalloidin; red), and nuclei (DAPI; blue) after 3 h of adhesion in a serum-free medium and imaged using confocal microscopy (60 magnification). Both were reproduced with permission from [148]. Copyright: Elsevier Ltd.

**Figure 11 pharmaceutics-15-00524-f011:**
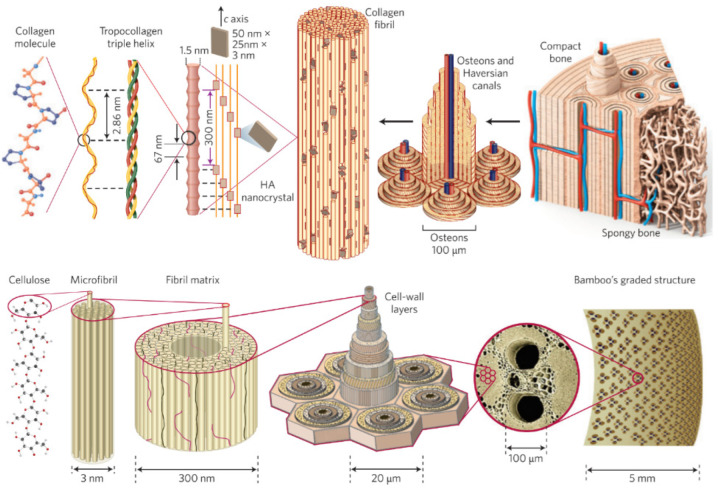
Comparison of the hierarchical structures of bone and bamboo. Reproduced with permission from [149]. Copyright: Nature Publishing Group.

**Figure 13 pharmaceutics-15-00524-f013:**
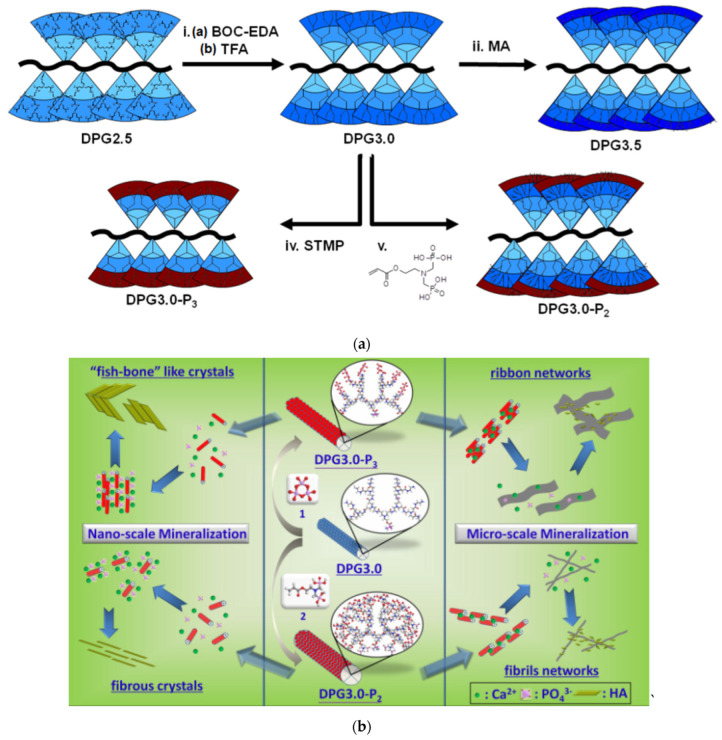
(**a**) Schematic illustration of the syntheses of dendronized PAMAMs (**b**) Phosphorylation of dendronized PAMAMs and their directed HA biomineralization on both nano- and micro-scales. Both were reproduced under permission from [170]. Copyright: ROYAL SOCIETY OF CHEMISTRY.

**Figure 14 pharmaceutics-15-00524-f014:**
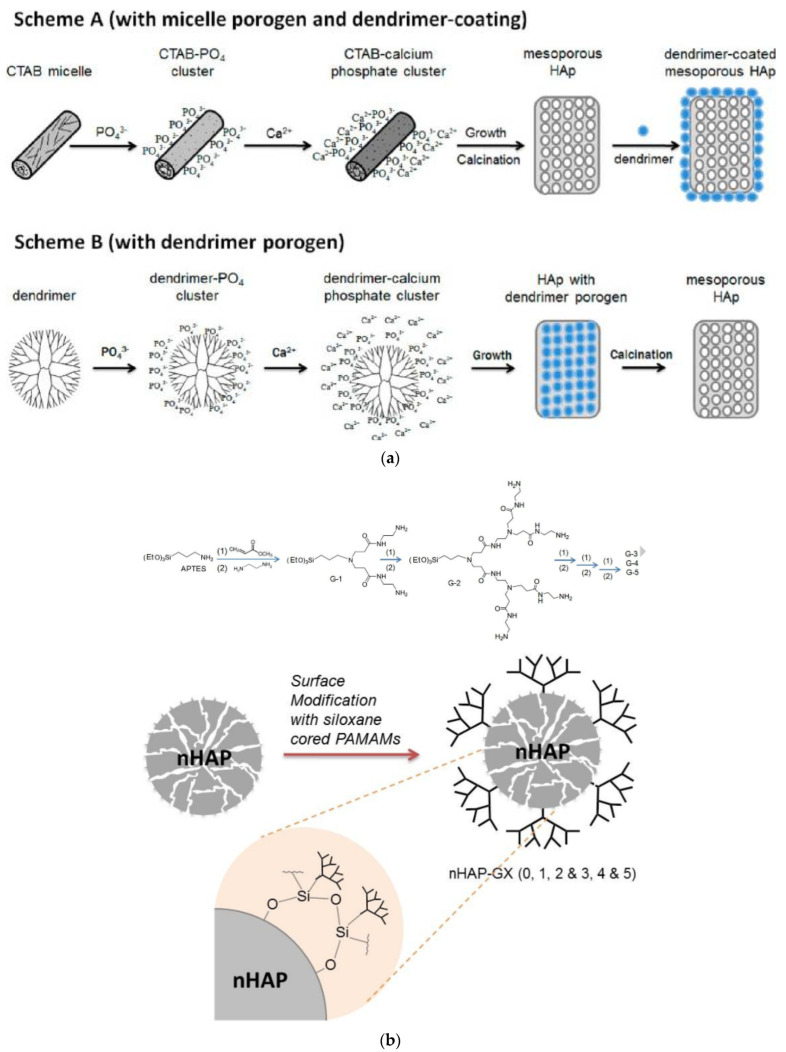
(**a**) Formation Process of Mesoporous HAp Powders Using Micelle and Dendrimer Porogens. Reproduced with permission from [172]. Copyright: American Chemical Society. (**b**) Synthetic path for the production of HAP functionalized by siloxane PAMAM dendrons Reproduced with permission from [173]. Copyright: Elsevier Inc.

**Figure 17 pharmaceutics-15-00524-f017:**
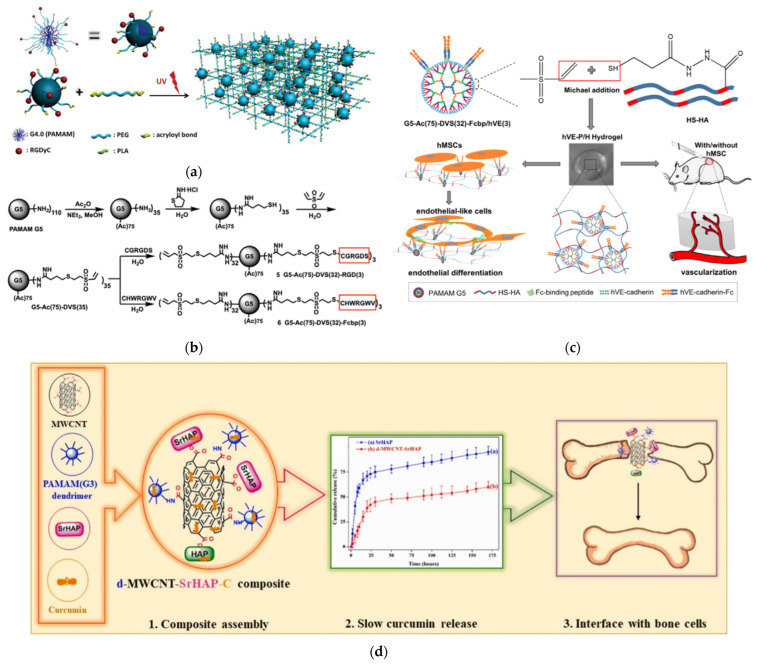
(**a**) Schematic representation of the hydrogel derived from functionalized dendritic G4 PAMAM and linear PEG. Reproduced with permission from [194]. Copyright: WILEY-VCH Verlag GmbH & Co. KGaA, Weinheim (**b**) Synthesis of the functionalized PAMAM dendrimer. (**c**) Preparation of the G5 PAMAM-thiolated hyaluronic acid hydrogel and endothelial differentiation of human umbilical cord mesenchymal stem cells and vascularization in vitro and in vivo. Both were reproduced with permission from [196]. Copyright: Elsevier Ltd. (**d**) Schematic representation of 1. composite assembly. 2. Curcumin release from the composite in comparison to strontium-substituted HAP 3. Osseointegration of the hybrid material with the damaged bone. Reproduced with permission from [197]. Copyright: Elsevier Ltd. and Techna Group S.r.l.

**Figure 18 pharmaceutics-15-00524-f018:**
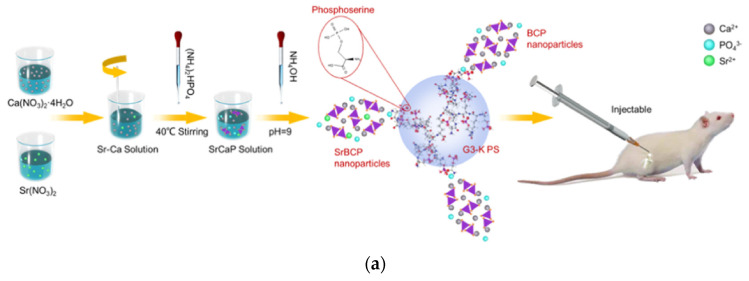
(**a**) Schematic diagram of the main steps of preparing bone injectable pastes for treating osteoporotic bone defect. (**b**) Reconstructed micro-computed tomography images of the coronal sections from the metaphyseal femur at week 12 (the red circles indicate new bone formation within a 3 mm diameter defect) and 3D reconstruction of the newly formed bone inside the defected area. Both were reproduced with permission from [203]. Copyright: American Chemical Society. (**c**) Schematic representation of the hydrogel from the aminoacid derivatives of the G4.0 PAMAM dendrimer and the PEG star polymer Reproduced with permission from [204]. Copyright: Royal Society of Chemistry.

**Figure 19 pharmaceutics-15-00524-f019:**
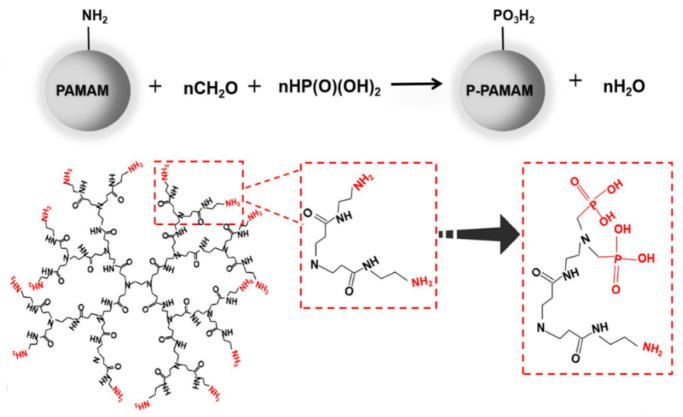
Synthesized mechanism of PAMAM-PO_3_H_2_ via the Mannich-type reaction Reproduced under permission from [208]. Copyright: The Author(s).

**Figure 20 pharmaceutics-15-00524-f020:**
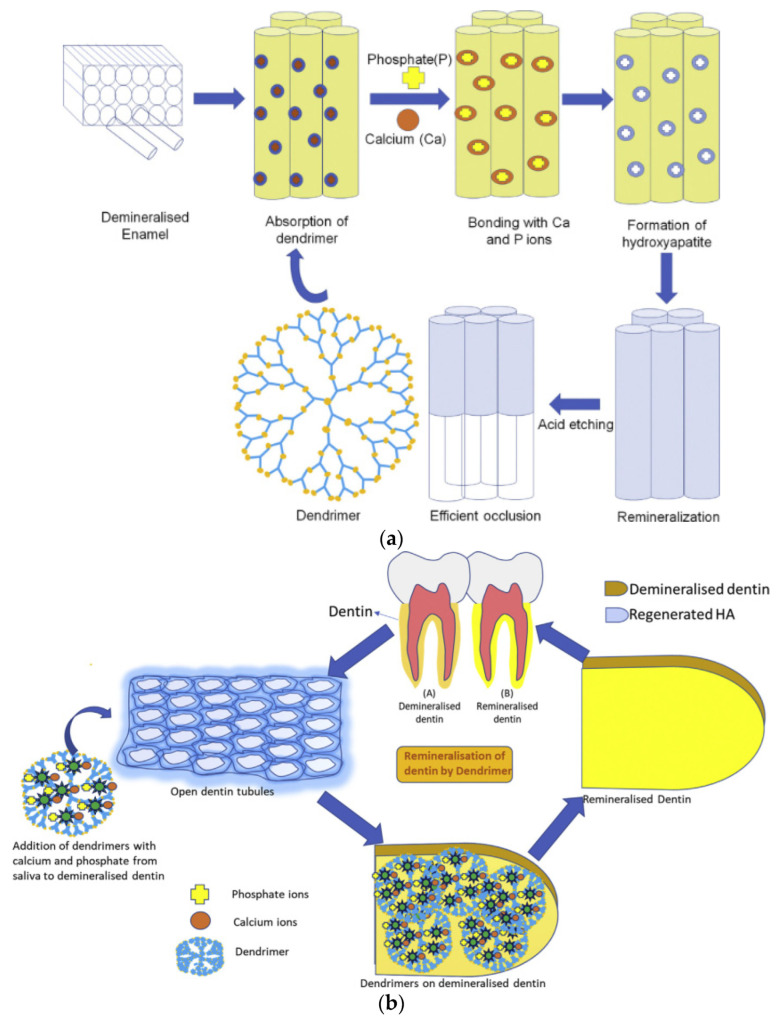
(**a**) Remineralization of demineralized enamel by dendrimers. (**b**) Remineralization of demineralized dentin by dendrimers. Both were reproduced under permission from [209]. Copyright: Elsevier Ltd.

**Figure 28 pharmaceutics-15-00524-f028:**
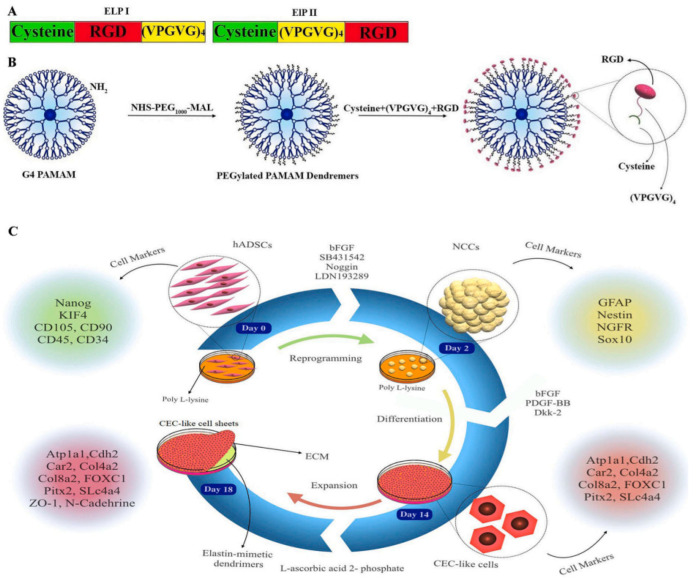
(**A**) Schematic representation of the amino acid sequences. (**B**) Schematic diagram for the synthesis of temperature-sensitive elastin mimetic dendrimers. (**C**) Stages of differentiation of hADSCs into CEC-like cells. Finally, the CEC-like cell sheet formation on thermo-responsive elastin-mimetic dendrimers. The growth factors and specific markers used at each stage of the differentiation protocol are shown. Published under Creative Commons CC-BY license from [269].

**Figure 29 pharmaceutics-15-00524-f029:**
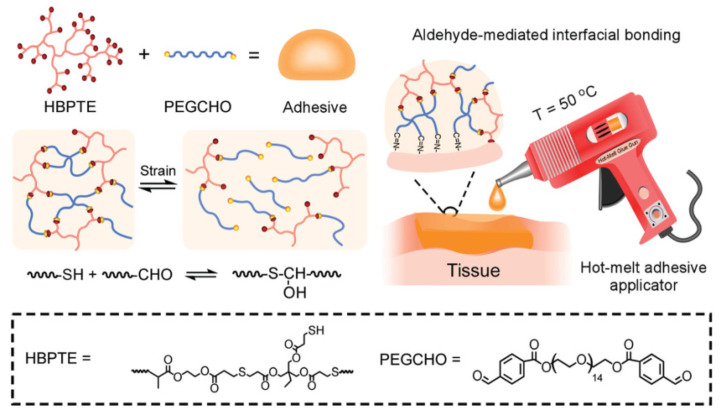
Schematic illustration of the fabrication of solvent-free adhesive using the dynamic thiol–aldehyde crosslinking chemistry and its administration using a hot-melt adhesive applicator. Reproduced with permission from [272]. Copyright: Royal Society of Chemistry.

**Figure 30 pharmaceutics-15-00524-f030:**
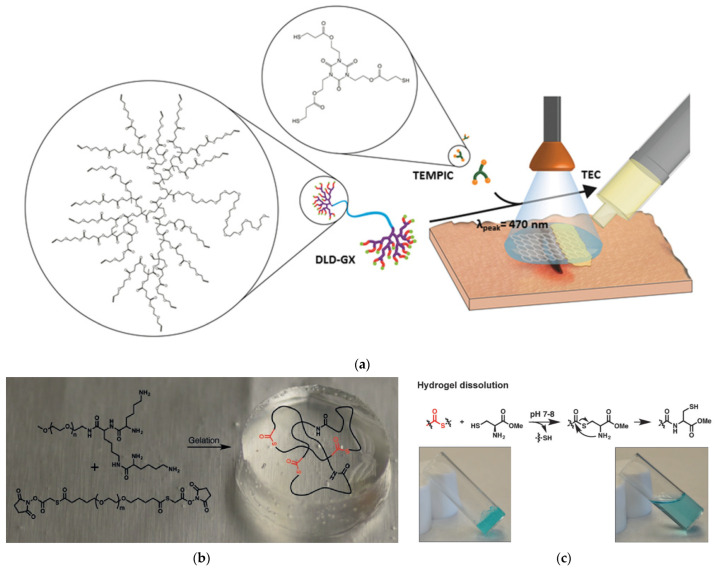
(**a**) Schematic representation of dendritic–linear–dendritic copolymer (DLD-GX) crosslinked with tris [2-(3-mercapto propionyl oxy)ethyl] isocyanurate (TEMPIC). The soft tissue adhesive patch is easily applied and gently cured through thiol–ene coupling (TEC) upon light initiation with HEV to stabilize a soft tissue wound. Reproduced with permission from [273]. (**b**) Lysine dendron gelation with the PEG crosslinker incorporating the thiolate groups; (**c**) Hydrogel dissolution reaction based on a thiol–thioester exchange. Both were reproduced under permission from [274]. Copyright: WILEY-VCH Verlag GmbH & Co. KGaA, Weinheim.

**Figure 31 pharmaceutics-15-00524-f031:**
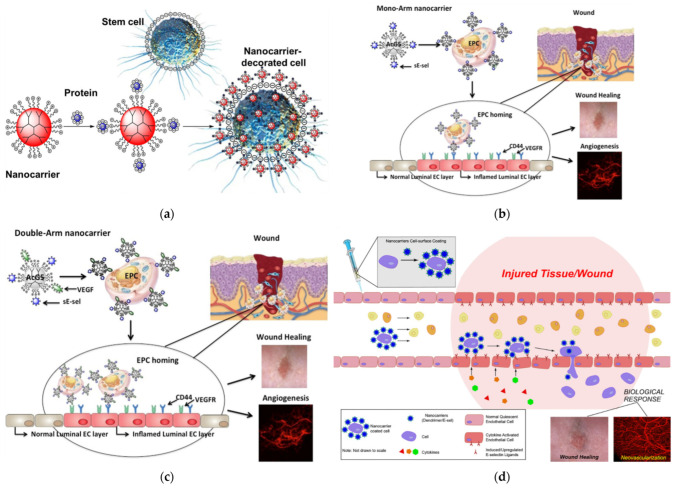
(**a**) Schematic illustration for the design of a nanocarrier for cell coating; Representative scheme of the dendritic polymer functionalization for the desired (**b**) “Mono-arm” biological readout. (Ac-G5-PAMAM-sE-sel) and (**c**) “double-arm” biological readout. (Ac-G5-PAMAM-sE-sel/VEGF) for endothelial progenitor cell coating. (**d**) Illustration of this novel targeted systemic cell delivery system. The cell surface is coated with nanocarriers composed of dendrimers conjugated with adhesion molecules. These nanocarriers guide coated cells homing to the desired tissue via association with the counterpart molecules highly or selectively expressed on the endothelium of diseased tissues. published under Creative Commons CC-BY license from [277].

**Figure 32 pharmaceutics-15-00524-f032:**
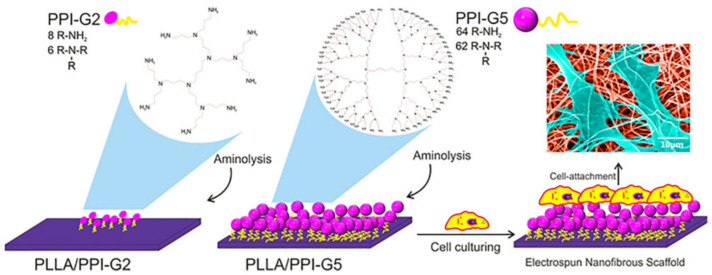
Incorporation of the second (G2) and fifth generation (G5) of Poly (propylene imine) (PPI) dendrimers onto the surface of PLLA scaffold via aminolysis reaction and the cultured fibroblast cells on the surface of electrospun PLLA/PPI-G5 scaffold. Reproduced with permission from [279] Copyright: Taylor & Francis.

**Figure 33 pharmaceutics-15-00524-f033:**
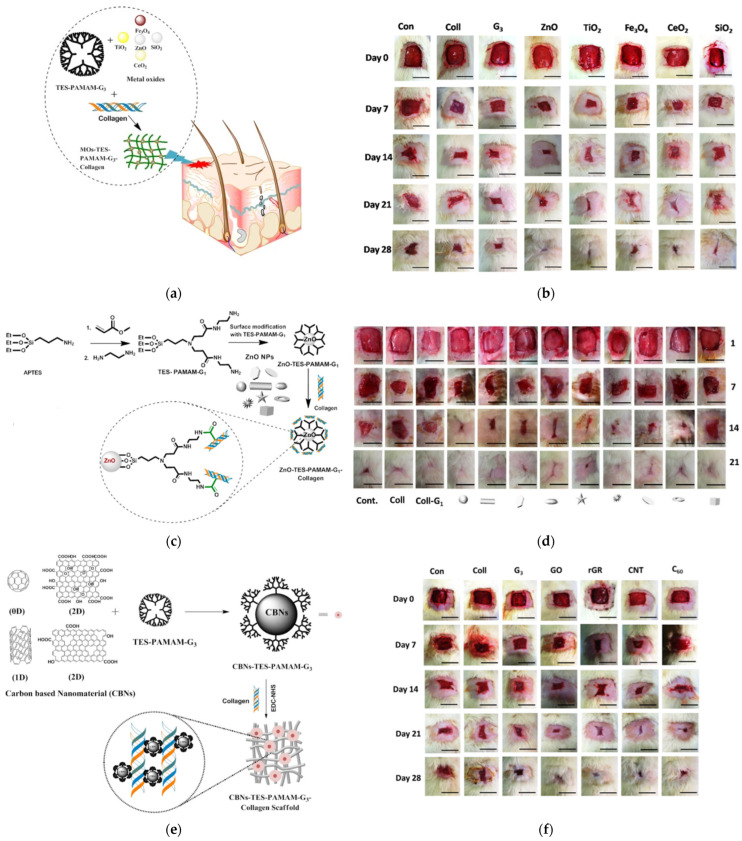
(**a**) Schematic diagram of the construction of collagen scaffolds (**b**) Representative photographs of excisional wounds on Wistar albino rats at different time intervals following treatment with ZnO, TiO_2_, Fe_3_O_4_, CeO_2_, and SiO_2_. Scale bar: 5 mm. Both are published with permission from [280]. Copyright: Springer Nature. (**c**) Schematic diagram depicting the method of combining different shaped ZnO-TES-PAMAM-G1 Nps with collagen scaffold. (**d**) The wound healing capacity of collagen, collagen-TES-PAMAM-G1, and variable morphology of ZnO nanoparticle cross-linked collagen scaffold on Wistar Albino rats at days 1, 7, 14, and 21. Both are published with permission from [281]. Copyright: American Chemical Society. (**e**) Schematic representation of the production of cross-linked collagen scaffold with G3 PAMAM functionalized carbon nanomaterials. (**f**) Representative photographs of full-thickness excision wound healing process. Both are published with permission from [282]. Copyright: Elsevier Ltd.

**Figure 34 pharmaceutics-15-00524-f034:**
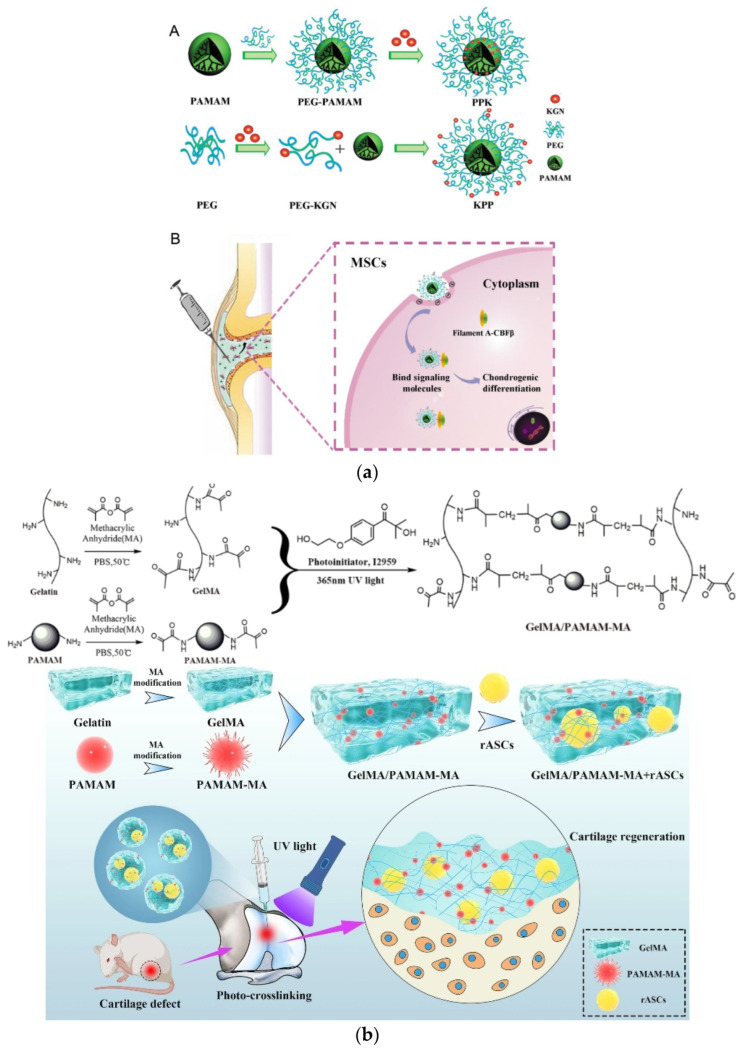
(**a**) Schematic illustrations of construction (**A**) and chondrogenic differentiation effect (**B**) of KGN-PAMAM conjugates. Through IA injection, KGN-PAMAM conjugates with positive charge were taken up by the mesenchymal cells via adsorptive endocytosis and then delivered to the cytoplasm where KGN competitively interacted with signaling molecules and thus exerted chondrogenic differentiation effects [286]. Copyright: Elsevier Inc. (**b**) Synthesis of photo-crosslinked GelMA/PAMAM-MA hydrogel and schematic diagram of the injectable stem cell-laden photo-crosslinked GelMA/PAMAM-MA hydrogel for cartilage regeneration Published under Creative Commons License from [287]. Copyright: The Author(s).

**Table 1 pharmaceutics-15-00524-t001:** The involvement of dendritic polymers in Collagen and Elastin biomimicry.

Dendritic Polymer	Modification	Function	Reference
Trimesic acid core	Gly-Pro-Nleu and Gly-Nleu-Pro sequences as branches	Collagen Mimetic	[119]
G1-G3 PPI	-	Collagen Crosslinking	[124]
G0.5 PAMAM core	Gly-Pro-Nleu sequence as branches	Collagen Mimetic	[120]
G3.5 PAMAM	collagen model peptide, (Pro-Pro-Gly)_5_	Collagen Mimetic	[121]
G4 PAMAM	(Pro-Pro-Gly)_10_	Collagen Mimetic	[122]
G4 PAMAM	(Pro-Hyp-Gly)_10_	Collagen Mimetic Hydrogel	[123]
G4 PAMAM	Ac-Val-Pro-Gly-Val-Gly	Elastin Mimetic	[125]
G3, G4, G5 PAMAM	Val-Pro-Gly-Val-Gly	Elastin Mimetic	[126]
G4 PAMAM	Val-Pro-Gly-Val-Gly Au Nps	Elastin Mimetic	[127]

**Table 2 pharmaceutics-15-00524-t002:** Dendritic Polymers that promote cell adhesion, organization, proliferation, and differentiation.

Dendritic Polymer	Modification	Substrate	Cell Type	Ref.
G1 PAMAM	-	Collagen (cholecyst)	Murine 3T3 fibroblasts	[129]
G2 PAMAM	-	Collagen	Human conjunctival fibroblasts	[130]
G2 PPI	-	Bovine type-I collagen, human collagen (HC), and human extracellular matrix (hECM)	3T3 fibroblast cell	[132]
dPAMAM transfection reagent	plasmid DNA	collagen-chondroitin sulfate scaffold	Mesenchymal stem cells	[113]
G1 PAMAM	pro-survival peptide analogs	Collagen	Stem cells	[131]
G5 PAMAM	Rhodamine	poly-lactide (PLA), miRNA mimics (premir-503)	HeLa Cells	[135]
G2 PAMAM	betamethasone sodium phosphate	Gelatin methacrylate	L929 cell line, anti-inflammatory	[136]
G2 PAMAM		Silk fibroin nanofibers	Fibroblasts cell line L929	[134]
G2 PAMAM	collagen-mimetic peptides, cell-binding sequence enzymatic cross-linking	Cell Aggregate	Hepatoma-liver	[148]
Dendrimer hydrazides	multi-cellular aggregates	Cell Aggregate	Human hepatoblastoma HepG2 cell line	[145]
G3 PAMAM	PEG acrylate photo–cross-linking	Hydrogel	RAW264.7 macrophages	[140]
G3 PPI hexadecaamine	Oleyl- polyethylene glycol	Cell Aggregate	C3A cells	[144]
Hyperbranched polyesters	Methacrylation photo–cross-linking	Hydrogel	NIH-3T3 fibroblasts	[141]
G4 PAMAM	RGD peptide	Cell Aggregate	NIH-3T3 fibroblasts	[146]
Hyperbranched PG	Phenol Groups Glucose	Hydrogel	Living Fibroblast cells	[142]
G3 PAMAM	Arg-Gly-Asp and PEG spacer	Cell Aggregate	Human hepatoblastoma HepG2 cell line	[147]

**Table 3 pharmaceutics-15-00524-t003:** Dendritic polymers’ contribution to bone regeneration.

Dendritic Polymer	Modification	Substrate	Function	Ref.
3.5, 4.5 PAMAM	-	-	Hydroxyapatite hemoglobin aquasomes	[154]
PPI	SDS	-	Hydroxyapatite nucleation	[155]
Dendritic Poly(L-lysine) Star Polycaprolactone	-	Hydroxyapatite	MG63 Cell Proliferation	[175]
G5.5 PAMAM	-	-	Hydroxyapatite nucleation	[159]
G1 to G4 PAMAM		-	Hydroxyapatite nucleation	[140]
PAMAM	Carboxylic/polyhydroxy terminated	-	Hydroxyapatite nucleation	[139]
G1.5 PAMAM	Dexamethasone	Carboxymethyl chitosan	Human osteoblast-like cells/Rat bone marrow stromal cells proliferation	[187]
PEI 25000	-	-	Calcium phosphate nucleation	[165]
G1.5 PAMAM	Dexamethasone carboxymethyl chitosan	Hydroxyapatite or starch–polycaprolactone scaffolds	Osteogenic differentiation rat Bone marrow stromal cells	[188]
G5, G6, G7 PAMAM	-	-	Mesenchymal stem cells osteogenic differentiation	[184]
G1.5 PAMAM	dexamethasone carboxymethyl chitosan	hydroxyapatite or starch–polycaprolactone scaffolds	Rat bone marrow stromal cells proliferation	[189]
G4 PPI	-	-	Hydroxyapatite nucleation	[162]
G5 PAMAM	PEG	Ti	human bone mesenchymal stem cells proliferation	[174]
PEI 25000		Ti	Hydroxyapatite nucleation	[166]
G2 PAMAM	Glutamic Acid	Gelatin Gel	Hydroxyapatite nucleation	[169]
G1.5 PAMAM	Carboxymethyl chitosan, dexamethasone	-	rat bone marrow stromal cells differentiation to osteoblasts	[190]
G4 PAMAM		Hydroxyapatite	Hydroxyapatite nucleation	[172]
Poly(1-lysine) dendrons	Phosphoserine	Etched titanium oxide	Calcium phosphate nucleation MG63 and SAOS-2 osteoblast-like cells proliferation	[191]
G4 PPI		Ti	Hydroxyapatite nucleation	[167]
G2, G3 PPI		MWCNTs	human osteosarcoma (MG-63) cell line	[156]
G4 PAMAM	PEG, Arg-Gly-Asp-(D-Tyr)-cysteine (RGDyC)	poly(lactic acid)-b-poly(ethylene glycol)-b-poly(lactic acid)	Mouse bone marrow mesenchymal stem cells differentiation for bone formation	[194]
Dendronized PAMAM	tri-phosphate or bis-phosphonate peripheral groups		Hydroxyapatite nucleation	[170]
Poly(Epsilon-Lysine) Hyperbranched Dendrons	Phosphoserine	Calcium Phosphate Gels	Mesenchymal stem cell differentiation for bone cell stimulation	[193]
G2, G4, G6 PAMAM-succinic acid 1,12-diaminododecane core	Gold Nanoparticles	-	Hydroxyapatite nucleation	[157]
Poly(epsilon-lysine) dendrons	Phosphoserine	Ti	Murine osteoblastic MC3T3 cells and primary bone marrow cells proliferation	[192]
Bifunctional dendrimers with cyclotriphosphazene core	Thioctic acid	Gold surfaces	Proliferation of human osteoblasts	[179]
G5 PAMAM	Arg-Gly-Asp, Hyaluronic acid	-	Bone marrow stem cells proliferation	[180]
Poly(Epsilon-Lysine) Hyperbranched Dendrons	phosphoserine	Titanium implants	Osseointegration	[200]
G4.0 PAMAM	Maleimide-PEG-succinimidyl carbonate ester, methoxy-PEG-succinimidyl carbonate ester	3, 4-dihydroxy-L-phenylalanine DOPAD-terminated 8-armed PEG	Mouse bone marrow mesenchymal stem cells	[204]
G5 PAMAM	RGD, YIGSR, and IKVAV peptides	-	Bone marrow stem cells, pheochromocytoma (PC12) cells proliferation	[186]
Hyperbranched aliphatic polyester	Polythiophene, poly(ε-caprolactone)	-	Mouse osteoblast MC3T3-E1 proliferation	[182]
PEI 5000	Hydroxyapatite	Chitosan	In vivo bone formation	[163]
G4 PAMAM	Amino acids	Luciferase reporter plasmid DNA	Human adipose-derived mesenchymal stem cells (AD-MSCs) osteogenic-chondrogenic differentiation	[185]
G1 PAMAM	Arg-Gly-Asp tripeptides	Polystyrene	Human bone mesenchymal stem cells	[177]
G1-G5 PAMAM Dendrons	Triethoxysillyl focal point	-	Hydroxyapatite nucleation	[173]
Dendritic Amphiphile	cholesterol	-	Calcium Phosphate Mineralization	[158]
PEI 5000	Hydroxyapatite	Chitosan	In vivo bone formation	[198]
G1 PAMAM	Alginate	-	MC3T3-E1 pre-osteoblasts proliferation	[181]
G3 poly(epsilon-lysine) dendrons	Phosphoserine	Strontium-doped hydroxyapatite gel	Macrophage osteogenic differentiation, In vivo bone formation	[202]
G5 PAMAM	8arm-PEG-SH	-	Rat bone marrow mesenchymal stem cells osteogenic differentiation	[195]
PAMAM G1/G2/G3	carbon nanotubes (CNTs)	nanostructured hydroxyapatite	osteoblast-like MG 63 cell line	[183]
G5 PAMAM	CO_2_H- or CH_3_-NH2 groups	Silicon	MG-63 osteoblastic cells	[176]
G1.5, G2.5, G3.5 PAMAM	chitosan, ketoprofen		Hydroxyapatite	[164]
Amide-based amino terminal dendrons	Arg-Gly-Asp	Ti	Human osteoblastic cells	[178]
G3 poly(epsilon-lysine) dendron	phosphoserine	Biphasic calcium phosphate	Biomineralization and stem cell osteogenic differentiation bone marrow stromal cells proliferation	[203]
PEI 5000	Hydroxyapatite	Chitosan	In vivo bone formation	[199]
G3 PAMAM	Sr Hydroxyapatite, Curcumin	d-MWCNT	Human osteoblast-like MG-63 cells, proliferation, differentiation	[197]
G5 PAMAM	VE-cadherin	Thiolated hyaluronic acid	Human umbilical cord mesenchymal stem cells differentiation	[196]

**Table 4 pharmaceutics-15-00524-t004:** Dendritic polymers in Dentistry.

Dendritic Polymer	Modification	Substrate	Function	Ref.
G5 PAMAM	Arg-Gly Asp	-	Dermal microvessel endothelial, human vascular endothelial, odontoblast-like MDPC-23 cells	[206]
G5 PAMAM	Fluorescein isothiocyanate Arg-Gly Asp	-	Dental pulp cells differentiation mouse odontoblast-like MDPC-23 cells	[207]
G3, G4 PAMAM-COOH	Carboxylic acid functionalization	Collagen fibrils	Dentin remineralization	[210]
G3.5 PAMAM	Alendronate	Demineralized enamel	Enamel remineralization	[252]
G4 PAMAM-COOH	Butanedioic anhydride	Ferric chloride solution	Aggregates with a microribbon structure	[251]
PAMAM G3	Carboxylic acid functionalization	Demineralized enamel	Enamel remineralization	[245]
G1 PAMAM Dendron	Aspartic acid, hexadecyl chain at the focal point	Biomimetic amelogenin nanospheres	Hydroxyapatite orientation Enamel	[250]
G4 PAMAM	Dimethyl phosphate	Demineralized enamel	Enamel remineralization	[249]
G3 PAMAM-NH2		Demineralized dentin	Dentin remineralization	[217]
G4 PAMAM-COOH	Triclosan	Demineralized dentin	Dentin remineralization, anti-bacterial	[239]
G4 PAMAM		Demineralized dentinal tubules	Dentin remineralization	[215]
G3 PAMAM-NH2		Type-I collagen fibrils	Mineralization of collagen fibrils	[219]
G2 G4-PAMAM-OH		Demineralized dentin	Dentin remineralization	[212]
PAMAM-COOH	Ca(OH)_2_	Demineralized dentin	Dentin remineralization	[221]
G4.5 PAMAM-COOH		Demineralized enamel	Enamel remineralization	[246]
G2 PAMAM	Glutaraldehyde	Demineralized dentin	Dentin remineralization	[213]
G3 PAMAM	Phosphorylation polyacrylic acid	Demineralized dentin	Dentin remineralization	[233]
G4 PAMAM	Phosphorylation	Demineralized dentin	Dentin remineralization	[234]
G3 PAMAM	Amorphous calcium phosphate	Demineralized dentin	Dentin remineralization	[223]
G3.5 PAMAM		Demineralized dentin	In vivo Dentin remineralization	[211]
PAMAM-COOH	Nano-hydroxyapatite	Demineralized dentin	Dentin remineralization	[237]
PAMAM-OH, PAMAM-COOH, PAMAM-NH2		Demineralized dentin	Dentin remineralization	[218]
G3 PAMAM	Nanoparticles of amorphous calcium phosphate (NACP)	Demineralized dentin	Dentin remineralization	[225]
G3-PAMAM-NH2	Nanoparticles of amorphous calcium phosphate (NACP)	Demineralized dentin	Dentin remineralization in an acidic solution	[226]
PAMAM 3.5	NACP resin	Demineralized dentin	Dentin remineralization	[227]
PAMAM		Demineralized dentin	Dentin remineralization	[216]
G3 PAMAM	bioactive multifunctional composite	Demineralized dentin	Dentin remineralization	[230]
G3 PAMAM	Glutaraldehyde	Demineralized dentin	Dentin remineralization	[214]
G3 PAMAM-NH2	Amorphous calcium phosphate nanoparticles	Ethoxylated bisphenol A, dimethacrylate pyromellitic glycerol dimethacrylate resin	Dentin remineralization	[228]
Phosphorylated G3, G4 PAMAM	Apigenin	Demineralized dentin	Dentin remineralization, anti-bacterial	[240]
G4 PAMAM		Mesoporous bioactive glass nanoparticles	Dentin remineralization	[238]
G4 PAMAM	SN15 salivary statherin protein-inspired peptide NACP resin	Demineralized dentin	Enamel remineralization	[253]
PAMAM	Nanoparticles of amorphous calcium phosphate	Demineralized dentin	Dentin remineralization	[222]
G3 PAMAM-COOH	Nanoparticles of amorphous calcium phosphate	Demineralized dentin	Dentin remineralization	[224]
PAMAM-NH2, PAMAM-COOH, PAMAM-OH		Enamel	Enamel remineralization	[247]
G5 PAMAM-NH2, PAMAM-COOH		Demineralized enamel	Enamel remineralization	[248]
PAMAM-PO_3_H_2_		Demineralized dentin and type I collagen matrix	Hydroxyapatite, Dentin remineralization	[232]
G3 PAMAM	Pulpine	Demineralized dentin	Dentin remineralization, pulp tissue repair	[243]
PAMAM	Honokiol-loaded	Demineralized dentin	Enamel remineralization, anti-bacterial	[254]
G4 PAMAM-COOH	Chlorhexidine	Demineralized dentin	Dentin remineralization	[242]
G3 PAMAM-NH2	Nanoparticles of amorphous calcium phosphate, pyromellitic glycerol dimethacrylate, ethoxylated bisphenol A dimethacrylate resin	Demineralized dentin	Dentin remineralization in an acidic solution after severe fluid challenges	[229]
G4 PAMAM- PO_3_H_2_ G4 PAMAM-COOH		Demineralized dentin	Dentin biomimetic remineralization and dentinal tubule occlusion	[235]
G4 PAMAM-COOH	Amorphous calcium phosphate nanofillers, chlorhexidine	Demineralized dentin	Dentin remineralization, anti-bacterial	[241]
PAMAM- PO_3_H_2_			Dental pulp stem cells	[208]
PAMAM- PO_3_H_2_ PAMAM-COOH		Collagen fibrils	Mineralization of type I collagen fibrils	[236]
PAMAM–COOH	Nano-sized amorphous calcium phosphate	Collagen fibrils	Mineralization of dentin type I collagen fibrils	[220]

**Table 5 pharmaceutics-15-00524-t005:** Dendritic Polymers for the remediation of eyes.

Dendritic Polymer	Modification	Substrate	Function	Ref.
PEG core methacrylated poly(glycerol succinic acid)	Cross-linking	-	Cornea wound healing	[255]
Lysine-based peptide dendrons	Poly(ethylene glycol dialdehyde) cross-linking	-	Cataract Surgeries	[259]
G1-PGLSAMA)2-PEG: polyethylene glycol, succinic acid glycerol, methacrylic acid	methacrylate ester photo–cross-linking	-	Cornea wound healing	[256]
G1-PGLSAMA)2-PEG: polyethylene glycol, succinic acid glycerol, methacrylic acid	methacrylate ester photo–cross-linking	-	Cornea wound healing	[257]
G1-PGLSA-MA)2-PEG3400) poly(ethylene glycol), succinic acid, glycerol, methacrylic acid, cysteine dendron	methacrylate ester photo–cross-linking, Self-Gelling	-	securing laser in situ keratomileuses (LASIK) flaps	[260]
Cysteine-terminated lysine dendron	PEG diester-aldehyde-pseudo proline peptide ligation	-	Corneal transplant	[251]
G2 PPI		Collagen	Human corneal epithelial cells proliferation	[265]
G2 PPI	YIGSR cell adhesion peptide	Collagen	Human corneal epithelial cells proliferation	[268]
G2 PPI	Fibroblast growth factor, heparin	Collagen	Cornea wound healing	[266]
Lysine cysteine dendritic polymer	PEG-butyric dialdehyde	Hydrogel	Sealing of clear corneal cataract incisions	[261]
PEG core dendritic polyesters	Methacryliation	-	Corneal laceration repair	[258]
Lys3Cys4 dendritic cross-linker	Propionaldehyde, butyraldehyde, 2-oxoethyl succinate PEG		Cornea wound healing	[262]
Poly(ethylene glycol) core succinic acid and glycerol branches	Cell adhesion ligand RGD	2-hydroxyethyl methacrylate hydrogel type I collagen	Corneal fibroblasts, corneal epithelial cell proliferation	[264]
G2 PPI	Heparin-binding fibroblast growth factor, heparin	Collagen	Corneal wound healing-human corneal epithelial cells proliferation	[267]
G4 PAMAM	Val-Pro-Gly-Val-Gly		Human adipose-derived mesenchymal stem cells differentiation to Human Corneal Endothelial Cells	[269]

**Table 6 pharmaceutics-15-00524-t006:** Dendritic polymers for healing wounds on the skin.

Dendritic Polymer	Modification	Substrate	Function	Ref.
Dendritic Thioester	PEG cross-linking	-	Wound closure	[271]
G3-G6 PG core hyperbranched 2,2-bis(hydroxymethyl) propionic acid	Tris [2-(3-mercapto propionyl oxy)ethyl] isocyanurate	Surgical mesh	Soft tissue adhesive patches	[273]
Pegylated lysine dendron	Poly(ethylene glycol disuccinimidyl valerate)	Hydrogel	Wound closure	[275]
Lysine-based dendron	Thioester with PEG cross-linker	-	Burn Wounds	[274]
G5 PAMAM	Acetylation and conjugation with protein	Stem cell	Wound closure	[277]
Peptide (arginine-lysine) polycationic dendrimers	-	Progenitor cell biological bandages	Human umbilical vein endothelial cells proliferation wound healing, antibacterial	[276]
G4 PAMAM	-	Gelatin scaffolds	Keratinocytes, fibroblasts proliferation dermal skin tissue engineering	[278]
G1 PAMAM silylated dendron	Collagen	ZnO nanoparticles	Human Epidermal Keratinocytes Wound healing	[281]
G3 PAMAM	Triethoxysillyl focal point	Graphene oxide, reduced graphene, carbon nanotube, and fullerene	Collagen wound closure	[282]
Poly(L-lysine) dendrigrafts	PEG	-	Human dermal fibroblasts proliferation	[270]
G3, G5 PAMAM, G2, G3, G5 PPI	-	Poly (L-lactic acid) scaffolds	Human dermal fibroblast cells, higher attachment, and homogeneous spread morphology	[279]
Hyperbranched polymer with thiol end groups	Benzaldehyde-terminated PEG (PEGCHO)	-	Wound Closure	[272]
G3 PAMAM	Triethoxysillyl focal point	ZnO, TiO_2_, Fe_3_O_4_, CeO_2_, and SiO_2_	Collagen wound closure	[280]

## Data Availability

Not applicable.

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
