# Peer review of "Dendritic Polymers in Tissue Engineering: Contributions of PAMAM, PPI PEG and PEI to Injury Restoration and Bioactive Scaffold Evolution"

_pharmaceutics, 2023, doi:10.3390/pharmaceutics15020524_

Round 1
Reviewer 1 Report
Nice and comprehensive review of the applications of dendrimers in the area of tissue engineering.
Author Response
Nice and comprehensive review of the applications of dendrimers in the area of tissue engineering.
We thank the reviewer for the kind and encouraging comment
Reviewer 2 Report
Pharmaceutics-2112512
Title: Dendritic Polymers Applications in Tissue Engineering
Overview
The article is quite interesting and I learned a lot from it. I am not an expert in the polymer synthesis or structure, but in their in vitro or in vivo tissue engineering applications. I did the assessment with that view. I hope to be able to contribute this review paper. Below are my comments and suggestions for the authors to evaluate. I think it's worth an effort to have the figures on one page only. This makes compression easier. But, as I said, I liked the text and leave suggestions to the authors.
Title
The title seems broad in relation to the text, most of the text talks about scaffolds based on PAMAM, PPI or PEG. I wonder if it's worth putting that in the title.
Introduction
I liked the introduction. But I confess that I found the text some far-fetched. I apologize if it was my lack of understanding of the chosen approach or lack of understanding of the subject. Was not donehistorical approach (as they were discovered), but a functional approach to the theme (in the first paragraph, almost philosophical). Still, many classic articles are cited. In this sense, it might be worthwhile, at the discretion of the authors, to use more recent texts in the introduction.
Minor suggestions
- Line 29 of the introduction: the authors use the term “cells of the microcosm”. It got weird for me. Simply “microscopics” perhaps?
- Line 34: “starburst” maybe you can put it in quotes.
- On page 2, line 50, first paragraph: it failed to say what a “Dendrimer” is.
- On page 2, line 70: the authors use “guests”. Would it be the best word to refer to the polymers that are associated with the main chain? In the same vein, the authors say “Biological entities”. Wouldn't they be better "biomolecules". Perhaps it is worth evading the discussion of what a "biological entity" is; for some it is something alive, for other beings who are aware that they are alive.
- On page 3, line 87, and elsewhere in the text, the authors use the term “chapter”. This is not a book chapter, although it seems. Perhaps Items or Sessions will be better.
3. Dendritic Polymers Mediating to Cell Adhesion Proliferation and Differentiation
I think there's a comma missing in the session title.
- On page 6, lines 150-151, it is written: “Fibroblasts were a common and the most convenient test subject since they can readily differentiate and adapt to the desired activity”. I think I know what the authors meant, but “differentiating fibroblasts” is a complex matter. Perhaps “adapting” or “responding” quickly to the polymer would be better.
- On line 153, “scaffold” instead of “framework”?
From 3.3 (Dendritic Glues for Cell Aggregation) the authors begin to describe results related to applications. I understand that it is more familiar for authors to talk about the synthesis and aggregation of materials. But the part that talks about the cellular response needs to be described a little better. For example, on page 7, from line 239, the authors say “Cellular functionalities were improved”. Best at what? What is the cell morphology? How does it increase functionality? Which parameter used? Some data are missing to make this clearer to the reader. This is repeated throughout the text.
4. Bone Restoration
The authors say “presents many similarities with bamboo” (line 273). I confess that I consider the comparison inappropriate, which perhaps was made thinking about long bones, which are hollow. But flat bones have a different architecture.
4.2 Osseointegration
Going back to the description of the in vitro/in vivo results.
- On page 18, line 397-398, it says "suitable substrate for the immobilization of MG63 osteoblast-like cells", are we talking about how long the adhesion takes? This information is important. How long to cultivate? This is important information for us.
- At several other points in the text, the authors say things like “proliferation increase”, by how much? How long to cultivate? Or “efficiency has been increased”, but by how much? About what? The “cells proliferate on the...”, proliferation for how long? Accelerating by how much? Facilitating how? Exceptional how? Better compared to what? This information is very important for in vitro and in vivo evaluation. It is worth a supplementary effort.
4.3 Bone Formation
The same suggestion for item 4.3. How much bone formed? How long implant? Were the data obtained by histological analysis? MicroCT? NMR? The concept of "bone-like" may vary depending on the form of analysis.
5. Applications of Dendritic Polymers in Dentistry
On page 26, lines 553-554, the authors say: “unexpectedly, for no apparent reason, the research path in this field followed the inverse direction: From materials that interact with cells to inorganic component reconstitution”. In fact, there is an explanation. Unlike the bone, for the formation of enamel, the tooth does not have the cell equivalent to the osteoblast that forms the organic matrix and mineralizes it (in this case, the ameloblast, which dies with the eruption of the tooth). Remembering that enamel has approximately 97% mineral; 2% water, 1% organic material. So the strategies are different.
In this section, and elsewhere in the text, the authors say “animal studies” or “in vivo studies”. But which animal model? In mice, rats, rabbits? Results and goals vary by model. Important to say the animal model is about.
On page 29, between lines 663-665. How much study time? How many patients?
6. Treatment of Eye Related Conditions
On page 35, between lines 771 and 773, are these results in animals (which one) or in humans? How long implant?
7. Skin
Page 38 line 834, the authors say “relevant to cornea restoration”. I got confused. Are you talking about the stratum corneum, or dermis, or skin? Or was it a spelling error, as it referred to the eye just before? The same thing will happen between lines 884-885.
8. Articular Cartilage Tissue Repair
Page 44 line 934 appears again “corneal tissue reparation” which for cartilage does not make sense.
An important point in this session. Which cartilage (hyaline, elastic or fibrous) are you talking about? This is very important as the ECM composition is different and therefore the significance of collagen production varies. It is important to always say which cartilage the implant or construct is in.
9. Hepatic Aortic Neural and Pancreatic Tissue Engineering
Page 45 line 989, says “... such as corneal fibroblasts”. The text refers to the liver and in figure 35 it says “HEP G2 cells”. It's worth checking. Adjust the subtitle?
References
- Approximately 30% of articles up to 5 years old and 20% of articles up to 3 years old. Reasonably updated. But this is a review, so I understand how appropriate.
- Pay attention to the format of the references, which are different from that recommended by the MDPI.
I think a list of abbreviations might be interesting after Conclusons.
Author Response
We thank the reviewer for his constructive comments and all the time he dedicated to ameliorate our work
I think it's worth an effort to have the figures on one page only. This makes compression easier
We agree with the reviewer and we modified the pagination accordingly.
The title seems broad in relation to the text, most of the text talks about scaffolds based on PAMAM, PPI or PEG. I wonder if it's worth putting that in the title.
The title was indeed broad and we consider the reviewer’s proposition a very good idea. We modified the title accordingly.
it might be worthwhile, at the discretion of the authors, to use more recent texts in the introduction
A number of recent references have been added I the introduction
Line 29 of the introduction: the authors use the term “cells of the microcosm”. It got weird for me. Simply “microscopics” perhaps?
We were referring to a specific class of dendritic cells. Since this long sentence confused, we rephrased it.
Line 34: “starburst” maybe you can put it in quotes
Corrected
- On page 2, line 50, first paragraph: it failed to say what a “Dendrimer” is.
A more clear definition of dendrimers is now provided.
- On page 2, line 70: the authors use “guests”. Would it be the best word to refer to the polymers that are associated with the main chain? In the same vein, the authors say “Biological entities”. Wouldn't they be better "biomolecules". Perhaps it is worth evading the discussion of what a "biological entity" is; for some it is something alive, for other beings who are aware that they are alive.
The authors are chemists and help from biologists is warmly welcome. We will eagerly adopt the term “biomolecules”. On the other hand, the term “guests” is typical for anything that is included in the dendritic cavities
- On page 3, line 87, and elsewhere in the text, the authors use the term “chapter”. This is not a book chapter, although it seems. Perhaps Items or Sessions will be better.
The term chapter is replaced by the term session throughout the whole manuscript.
- Dendritic Polymers Mediating to Cell Adhesion Proliferation and Differentiation
I think there's a comma missing in the session title.
Commas were added.
- On page 6, lines 150-151, it is written: “Fibroblasts were a common and the most convenient test subject since they can readily differentiate and adapt to the desired activity”. I think I know what the authors meant, but “differentiating fibroblasts” is a complex matter. Perhaps “adapting” or “responding” quickly to the polymer would be better.
The sentence was rephrased according to the suggestion of the reviewer
- On line 153, “scaffold” instead of “framework”?
Done
From 3.3 (Dendritic Glues for Cell Aggregation) the authors begin to describe results related to applications. I understand that it is more familiar for authors to talk about the synthesis and aggregation of materials. But the part that talks about the cellular response needs to be described a little better. For example, on page 7, from line 239, the authors say “Cellular functionalities were improved”. Best at what? What is the cell morphology? How does it increase functionality? Which parameter used? Some data are missing to make this clearer to the reader. This is repeated throughout the text.
Indeed, the paper is prepared by chemists for chemists advising on synthetic paths for obtaining materials for tissue engineering. Our scope though is to broaden the audience as much as possible. The description has been extended as advised.
- Bone Restoration
The authors say “presents many similarities with bamboo” (line 273). I confess that I consider the comparison inappropriate, which perhaps was made thinking about long bones, which are hollow. But flat bones have a different architecture.
The comparison and similarities are now limited to long hollow bones. The reviewer is invited to check analogous similarities between haemoglobin and chlorophyll
4.2 Osseointegration
- On page 18, line 397-398, it says "suitable substrate for the immobilization of MG63 osteoblast-like cells", are we talking about how long the adhesion takes? This information is important. How long to cultivate? This is important information for us.
The requested information was added to the text
- At several other points in the text, the authors say things like “proliferation increase”, by how much? How long to cultivate? Or “efficiency has been increased”, but by how much? About what? The “cells proliferate on the...”, proliferation for how long? Accelerating by how much? Facilitating how? Exceptional how? Better compared to what? This information is very important for in vitro and in vivo evaluation. It is worth a supplementary effort.
Incubation times and percentages of cell proliferation increase were added in the specific chapter and throughout the whole manuscript
4.3 Bone Formation
The same suggestion for item 4.3. How much bone formed? How long implant? Were the data obtained by histological analysis? MicroCT? NMR? The concept of "bone-like" may vary depending on the form of analysis.
Relevant information is now included in the text
- Applications of Dendritic Polymers in Dentistry
On page 26, lines 553-554, the authors say: “unexpectedly, for no apparent reason, the research path in this field followed the inverse direction: From materials that interact with cells to inorganic component reconstitution”. In fact, there is an explanation. Unlike the bone, for the formation of enamel, the tooth does not have the cell equivalent to the osteoblast that forms the organic matrix and mineralizes it (in this case, the ameloblast, which dies with the eruption of the tooth). Remembering that enamel has approximately 97% mineral; 2% water, 1% organic material. So the strategies are different.
We are thankful for the explanation offered by the reviewer. We incorporated it into the manuscript almost as provided as an honour.
In this section, and elsewhere in the text, the authors say “animal studies” or “in vivo studies”. But which animal model? In mice, rats, rabbits? Results and goals vary by model. Important to say the animal model is about.
Information on the animal models is added for the in vivo studies.
On page 29, between lines 663-665. How much study time? How many patients?
The requested information has been added to the text
- Treatment of Eye Related Conditions
On page 35, between lines 771 and 773, are these results in animals (which one) or in humans? How long implant?
In vivo experiments were performed on white leghorn chickens. The injured tissue was repaired in one minute
- Skin
Page 38 line 834, the authors say “relevant to cornea restoration”. I got confused. Are you talking about the stratum corneum, or dermis, or skin? Or was it a spelling error, as it referred to the eye just before? The same thing will happen between lines 884-885.
By the word relevant we wanted to highlight the similarity. It was a poor choice of a word that confused. We rephrased.
- Articular Cartilage Tissue Repair
Page 44 line 934 appears again “corneal tissue reparation” which for cartilage does not make sense.
We were referring to a dendric polymer that is commonly used for corneal tissue reparation. The sentence has been clarified
An important point in this session. Which cartilage (hyaline, elastic or fibrous) are you talking about? This is very important as the ECM composition is different and therefore the significance of collagen production varies. It is important to always say which cartilage the implant or construct is in.
It is hyaline cartilage. The information is incorporated into the text
- Hepatic Aortic Neural and Pancreatic Tissue Engineering
Page 45 line 989, says “... such as corneal fibroblasts”. The text refers to the liver and in figure 35 it says “HEP G2 cells”. It's worth checking. Adjust the subtitle?
The mention of corneal fibroblast was referring to a previous chapter. This is now clarified
References
- Approximately 30% of articles up to 5 years old and 20% of articles up to 3 years old. Reasonably updated. But this is a review, so I understand how appropriate.
- Pay attention to the format of the references, which are different from that recommended by the MDPI.
References will be automatically adjusted to the recommended format after the acceptance of the manuscript during the proofing process with the aid of the provided DOIs
I think a list of abbreviations might be interesting after Conclusions
This is a very good idea that we will eagerly adopt
Reviewer 3 Report
The abstract should be improved and extended
The same about the conclusion
“Gelatin a component of collagen…” it is simply wrong.
“Cytocompatibility was established against the L929 cell line [116].” Add more details.
References should be updated
More details () must be added to the presented examples in each section
There are many figures in the work. Remove some of them or move to supplementary
Author Response
We thank the reviewer for his time and useful comments
The abstract should be improved and extended
The abstract was extended and refined
The same about the conclusion
The conclusions section was extended as well
“Gelatin a component of collagen…” it is simply wrong.
The comment has been removed from the manuscript.
“Cytocompatibility was established against the L929 cell line [116].” Add more details.
A detailed description is now incorporated into the manuscript
References should be updated
A good number of recent citations have been added mainly in the introduction
More details () must be added to the presented examples in each section
More information about the animals in the in vivo experiments, the clinical trials, details on cell cultures, incubation times, cell morphology, and the overall advantages of dendrimer addition have been included in the text
There are many figures in the work. Remove some of them or move to supplementary
We had a request to add some figures on cell morphology and decided to take the middle path.